# Ambient 🧬 Proteins:
# Training Diffusion Models on Low Quality Structures

**Giannis Daras** [*]
CSAIL, MIT
gdaras@mit.edu

**Jeffrey Ouyang-Zhang** [*]
Computer Science, UT Austin
jozhang@utexas.edu

**Krithika Ravishankar**
Computer Science, UT Austin
krithravi@utexas.edu

**Costis Daskalakis**
CSAIL, MIT
costis@csail.mit.edu

**Adam Klivans**
Computer Science, UT Austin
klivans@cs.utexas.edu

**Daniel J. Diaz**
Computer Science, UT Austin
danny.diaz@utexas.edu

## Abstract

We present *Ambient Protein Diffusion*, a framework for training protein diffusion models that generates structures with unprecedented diversity and quality. State-of-the-art generative models are trained on computationally derived structures from AlphaFold2 (AF), as experimentally determined structures are relatively scarce. The resulting models are therefore limited by the quality of synthetic datasets. Since the accuracy of AF predictions degrades with increasing protein length and complexity, de novo generation of long, complex proteins remains challenging. Ambient Protein Diffusion overcomes this problem by treating low-confidence AF structures as corrupted data. Rather than simply filtering out low-quality AF structures, our method adjusts the diffusion objective for each structure based on its corruption level, allowing the model to learn from both high and low quality structures. Empirically, Ambient Protein Diffusion yields major improvements: on proteins with 700 residues, diversity increases from 45% to 86% from the previous state-of-the-art, and designability improves from 68% to 86%. All of our code, models and datasets are available under the following repository: https://github.com/jozhang97/ambient-proteins.

## 1 Introduction

Proteins are the fundamental building blocks of life. They accelerate chemical reactions by many orders of magnitude, convert sunlight into food, and underpin the myriads of processes within cells and organisms with the level of accuracy and precision required to sustain life [6, 32]. Unlike computational protein engineering—which focuses on improving the developability or function of existing proteins through computationally guided mutations for practical biotechnological applications [21, 40, 24, 41, 8, 36, 20, 52]—*de novo* protein design aims to create entirely new proteins with specified structures and functions, ultimately seeking to discover folds and activities not found in nature [12]. Since protein function is largely determined by tertiary and quaternary structure, generative machine learning frameworks for protein design focus on learning the sparse, evolutionarily sampled landscape of protein structures, with the goal of generating novel, functional backbone scaffolds beyond those observed in nature [50, 33, 34, 28, 23, 53, 22, 7, 55, 47].

Recent breakthroughs in machine learning–based structure prediction—most notably AlphaFold2 [29]—have made it possible to infer accurate protein structures directly from se-

---

[*]Equal contribution.

39th Conference on Neural Information Processing Systems (NeurIPS 2025).

quence [29, 11, 35]. This progress has enabled the creation of large-scale structural resources such as the AlphaFold Database (AFDB), which contains over 214M predicted structures from UniProtKB sequences [13, 46]. In parallel, high-throughput tools for sequence and structure comparison, such as MMSeqs2 and FoldSeek, have facilitated the curation of large, diverse training datasets from AFDB [5]. Among them, the 2.3M AFDB cluster dataset, has already been shown to improve the capabilities of generative models for protein structure design [34, 23].

The quality of a generative model depends on the size and fidelity of its training data. While AlphaFold2 (AF) has enabled large-scale protein structure prediction, its outputs often contain biological or computational inaccuracies [51]. To estimate the reliability of a predicted structure, AlphaFold provides a per-residue confidence score, the predicted Local Distance Difference Test (pLDDT), which is a proxy of local structural accuracy. In practice, researchers frequently filter predicted structures based on average pLDDT scores, training only on high-confidence subsets (typically using a cutoff of pLDDT $> 80$). However, lower pLDDT scores are disproportionately associated with longer and more structurally complex proteins. As a result, filtering based on pLDDT introduces a bias toward smaller, simpler folds, reducing structural diversity in the training set and impairing the model's ability to generalize to more complex regions of structure space—including longer proteins. Notably, many low-pLDDT structures still contain well-folded domains that are misoriented with respect to each other, as reflected by low predicted alignment error (pAE). These structures can still offer valuable domain-level and coarse-grained information about the structure distribution, which is discarded by overly aggressive filtering.

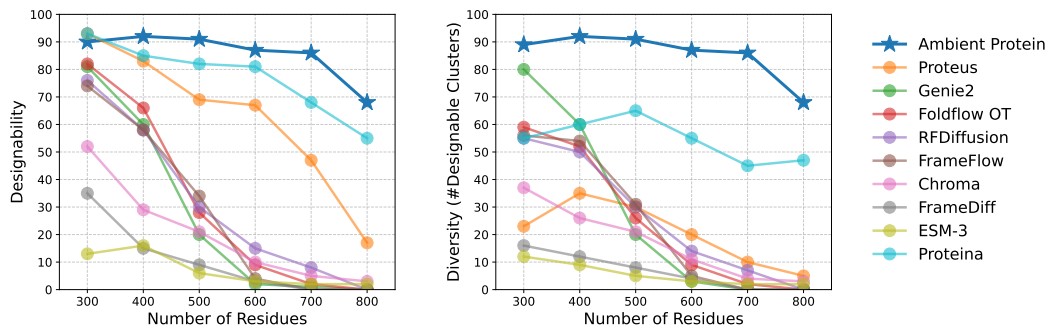

Figure 1: **Long protein generation performance.** We train Ambient Protein Diffusion on proteins up to 768 residues and sample sequences ranging from 300 to 800 residues. Our 17M parameters model significantly outperforms the previous state-of-the-art Proteína [23], which is a 200M parameters model. Ambient Protein Diffusion generates both diverse and designable structures across all lengths.

To mitigate these issues, we depart from the standard paradigm of aggressive filtering of low-confidence structures. Instead, we introduce *Ambient Protein Diffusion* —a framework for training diffusion models that incorporates proteins with noisy or incomplete structures directly into the training process. Ambient Protein Diffusion builds on recent advances in learning generative models from corrupted data [14, 16, 1, 15, 2, 42, 4, 49, 17, 37, 43, 27], which have explored controlled corruption settings such as additive Gaussian noise [14, 16, 1, 17] and masking [15, 2]. Our framework generalizes these techniques to arbitrary, unknown corruption processes, enabling the training of generative models in scientific domains where the corruption mechanism is complex and non-parametric. In our setting, the AlphaFold prediction errors represent such a corruption: they are structured, not explicitly modeled, and vary across protein size and topology. Yet, our method effectively leverages these imperfect samples to significantly advance the capabilities of generative protein models. For example, on proteins with 700 residues, our 16.7M parameter model improves diversity from 45% to 86% and increases designability from 68% to 86% compared to the previous state-of-the-art, Proteína [23], a 200M parameter model. Below, we summarize our key contributions:

- We generalize recent approaches for training generative models on corrupted data to handle arbitrary, non-parametric, and unknown corruption processes, enabling their application to scientific domains.

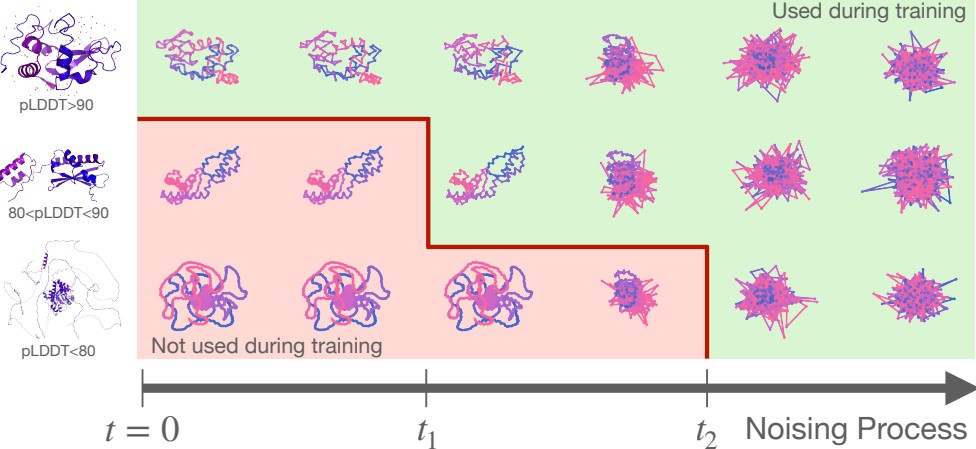

Figure 2: **Overview of Ambient Protein Diffusion on AlphaFold structures.** Rows 1-3 show the noising process (from left to the right) of three different AlphaFold proteins based on their average pLDDT (top: high, middle: medium, and bottom: low). These proteins are only used during training at the green diffusion times. At these noise levels, any initial AlphaFold prediction errors in low-pLDDT proteins have effectively been "erased" by the added noise, and the distributions of low- and high-pLDDT proteins have merged.

- We demonstrate that our framework, Ambient Protein Diffusion, effectively leverages low-pLDDT AlphaFold predictions, allowing the model to learn from all available samples without distorting the underlying structure distribution.

- We further construct a new training set from the AFDB cluster dataset optimized for geometric diversity irrespective of their evolutionary relationship, yielding a broader and more representative sampling of structural space for generative modeling.

- We achieve state-of-the-art results in both diversity and designability for protein generation, improve diversity by 45% and designability by 24% on long proteins (800 residues), and establish the Pareto frontier between these objectives on short proteins ($< 256$ residues). Our models also achieve state-of-the-art novelty scores in both short and long protein generation, indicating much lower memorization of the training set.

## 2 Background and Related Work

**De novo Protein Generation.** Most *de novo* protein generation frameworks that operate in structure space follow a three-step pipeline: (1) a generative model samples a three-dimensional backbone structure; (2) an inverse folding model (e.g., ProteinMPNN [19]) proposes amino acid sequences likely to fold into the generated backbone; and (3) these sequences are evaluated by a structure prediction model (e.g., ESMFold [35]) to identify the ones that best recapitulate the target fold.

Pioneering methods such as RFDiffusion [50] and Chroma [28] have established strong baselines for backbone generation. More recent advances include Genie [33], which introduces a denoising diffusion model with an SE(3)-equivariant network that generates proteins as point clouds of reference frames; Genie2 [34], which scales Genie using synthetic AlphaFold structures to improve training data diversity; and Proteína [23], which replaces diffusion with flow matching and scales both model size and dataset scale by orders of magnitude to improve performance on longer and more complex monomeric proteins.

Ambient Protein Diffusion is built using the Genie architecture and makes use of ambient protein diffusion to achieve state-of-the-art results with substantially shorter training times, much fewer parameters (16.7M vs 200M), and significantly reduced computational requirements.

**Training Datasets.** Recent advances in structure prediction—most notably AlphaFold2 [29] and ESMFold [35]—have dramatically expanded the available structural data, enabling the prediction

of $\sim$214M and $\sim$617M monomeric protein structures from UniProtKB (via the AlphaFold Protein Structure Database) [46] and metagenomic libraries (via the ESM Atlas) [35], respectively. While this explosion of computational structures presents unprecedented opportunities, it also poses significant challenges for downstream bioinformatic analysis and model training, particularly due to the scale, redundancy, and uneven quality of the predicted structures. To address this, prior work applied MMSeqs2 [26] and FoldSeek [45] to cluster the AlphaFold Database (AFDB), yielding $\sim$2.3M clusters shown to capture evolutionary relationships between predicted structures [5]. This AFDB cluster dataset has since served as the foundational dataset to train several generative protein structure models [23, 34].

In this work, we apply a reparameterized FoldSeek to the AFDB cluster dataset to maximize geometric diversity rather than evolutionary insights. Our goal is to construct a dataset better suited for learning a generative model of protein structure space—one that emphasizes structural rather than evolutionary variation. Starting from the 2.3 million AFDB clusters, we use the cluster representatives with average pLDDT > 70 ($\sim$1.29M structures) and apply our geometric clustering procedure. The resulting dataset comprises roughly $\sim$292K structurally diverse clusters.

**Diffusion Models.** The goal in diffusion modeling is to sample from an unknown density $p_0$ that we have samples available. Formally, let $\mathcal{D} = \{x_0^i\}_{i=1}^N$ a dataset of $N$ independent samples, where $X_0^i \sim p_0$. The unknown distribution $p_0$ is potentially complex, high-dimensional and multimodal. To make the sampling problem more tractable, in diffusion modeling we target smoothened densities $p_t$ defined as the convolution with a Gaussian: $p_t = p_0 * \mathcal{N}(0, \sigma^2(t)I_d)^2$, where $\sigma(t)$ is an increasing function of $t$, with $\sigma(0) = 0$. In particular, the object of interest in diffusion modeling is the score-function of the smoothened densities, defined as $\nabla_{x_t} \log p_t(x_t)$. The latter is connected to the optimal denoiser (in the $l_2$ sense) through Tweedie's Formula: $\nabla \log p_t(x_t) = \frac{\mathbb{E}[X_0|X_t=x_t]-x_t}{\sigma^2(t)}$.

Given access to $\mathbb{E}[X_0|X_t = x_t]$ one can sample from the distribution $p_0$ of interest by running a discretized version of a reverse diffusion process [3, 44, 9, 10]. Hence, the sampling problem becomes equivalent to the problem of approximating the set of functions $\{\mathbb{E}[X_0|X_t = \cdot]\}_{t=0}^T$. Given a sufficiently rich family of functions $\{h_\theta : \theta \in \Theta\}$, the conditional expectation at a particular time $t$ can be learned by minimizing the objective:

$$J(\theta) = \mathbb{E}_{t\in\mathcal{U}[0,T]}\mathbb{E}_{x_0,x_t|t}\left[||h_\theta(x_t) - x_0||^2\right]. \tag{1}$$

In the context of protein diffusion models for backbones, $X_0$ captures the 3-D co-ordinates for each residues of the protein.

**Learning from noisy data.** Recent work has explored the problem of learning diffusion models from corrupted data. Typically, the corruption process is simple, e.g. it can be additive Gaussian noise as in [14, 16, 1], or masking as in [1, 2]. Even in works where the corruption process is more general, the degradation needs to be known and multiple diffusion trainings are required until an Expectation-Maximization algorithm converges [42, 4]. In this work, we deviate from this setting as the corruption process is unknown and complex, which may include AlphaFold learning and hallucination errors, and noise inherent to the structural biology technique used to solve the structure, etc. We also target a single diffusion training instead of performing multiple EM iterations. The method is detailed in Section 3.

Our work generalizes the techniques developed in [14, 16] for the additive Gaussian noise case. Particularly, in [16], the authors consider learning from a dataset $\mathcal{D} = \{(x_{t_i}^i, t_i)\}_{i=1}^N$ of samples noised with additive Gaussian noise of different variances $\{\sigma^2(t_i)\}_{i=1}^N$. Formally, let $X_{t_i} = X_0 + \sigma(t_i)Z$, where $X_0 \sim p_0, Z \sim \mathcal{N}(0, I_d)$. Each point $X_{t_i}$ contributes to the learning only for $t \geq t_i$, using the objective:

$$\hat{J}(\theta) = \mathbb{E}_{t\in\mathcal{U}[0,T]} \sum_{x_{t_i} \in \mathcal{D}:\ t>t_i} \mathbb{E}_{x_t|x_{t_i},t_i} \left[||\alpha(t,t_i)h_\theta(x_t,t) + (1-\alpha(t,t_i))x_t - x_{t_i}||^2\right], \tag{2}$$

$\alpha(t,t_i) = \frac{\sigma^2(t)-\sigma^2(t_i)}{\sigma^2(t)}$. As the number of samples grows to infinity, Equation 2 also recovers the conditional expectation $\mathbb{E}[X_0|X_t = x_t]$, but it does so while being able to utilize noisy samples.

---

[2]Alternative formulations of diffusion modeling, such as the Variance Preserving case, are equivalent to this case up to a simple reparametrization. For the ease of analysis, we focus our presentation on corruptions of the form $X_t = X_0 + \sigma_t Z, \quad Z \sim \mathcal{N}(0, I_d)$.

This objective recovers the true minimizer because one can prove that the conditional expectation $\mathbb{E}[X_{t_i}|X_t = x_t]$, lies in the line that connects the current noisy point $x_t$ and the prediction of the clean image, $\mathbb{E}[X_0|X_t = x_t]$.

**Distribution merging under noise.** A key idea in our method will be that distribution distances contract as we increase the noise level added. In the context of diffusion models, this property has been leveraged in the SDEdit [39] paper to allow diffusion models to perform stroke-based editing at inference time without any finetuning. Most relevant to our work, Ambient Omni [18] uses the distribution contraction property (together with other innovations) to use blurry and out-of-distribution data during training. In this work, we focus on *synthetic data* from AlphaFold that arise from various corruption sources: biological ones (errors in the crystallography process), computational ones (errors in the solution of the inverse problem from the crystallography data to the modeled structure), and learning ones (AlphaFold mistakes due to the limited size of the training dataset and hallucinations).

## 3 Method

### 3.1 Building Intuition

We are given access to samples from the AlphaFold distribution $\tilde{p}_0$ and aim to learn how to sample from the true distribution of experimentally solved structures, $p_0$, without an explicit degradation model mapping $p_0 \rightarrow \tilde{p}_0$. Our key insight is that, regardless of how $\tilde{p}_0$ deviates from $p_0$, adding noise to both distributions causes them to contract toward one another. As the noise level increases, the distributions $\tilde{p}_t$ and $p_t$ become progressively more aligned. This is because it is known that Gaussian noise contracts distribution distances (KL divergence) in the following sense:

$$D_{\mathrm{KL}}(p_t||\tilde{p}_t) \leq D_{\mathrm{KL}}(p_{t'} \,||\, \tilde{p}_{t'}), \quad \forall t \geq t'. \tag{3}$$

In fact, as $t \rightarrow \infty$, we have that: $D_{\mathrm{KL}}(p_t \,||\, \tilde{p}_t) \rightarrow 0$, as both distributions converge to the same Gaussian. We now define the concept of merging of two distributions towards the same measure.

**Definition 3.1 ($\epsilon$-merged)** *We say that two distributions, $p$ and $\tilde{p}$ are $\epsilon$-merged, if the KL distance between the two is upper-bounded, by $\epsilon$, i.e., if $D_{\mathrm{KL}}(p \,||\, \tilde{p}) \leq \epsilon$.*

Similarly, we define the merging time of two distributions as the minimal amount of noise we need to add such that the two distributions become $\epsilon$-merged. Formally,

**Definition 3.2 ($\epsilon$-merging time)** *Let two distributions $p, \tilde{p}$. We define their $\epsilon$-merging time as follows:* $t_n(p, \tilde{p}, \epsilon) = \inf\{t : D_{\mathrm{KL}}(p * \mathcal{N}(0, \sigma(t)^2 I) \,||\, \tilde{p} * \mathcal{N}(0, \sigma(t)^2 I)) \leq \epsilon\}.$

Assuming we can estimate the $\epsilon$-merging time between two distributions $p$ and $\tilde{p}$, our key idea is to treat samples from $\tilde{p}_t$ as approximate samples from $p_t$ for all timesteps $t > t_n(p, \tilde{p}, \epsilon)$. This idea is illustrated in Figure 2. The intuition is that once the distributions have sufficiently merged under noise, the residual shift becomes negligible and samples from $\tilde{p}_t$ can be used for learning $p_t$. This holds because: (i) the learning algorithm may not be sensitive to small distributional discrepancies at high noise levels, and (ii) even if some bias is introduced, the remaining diffusion trajectory for times $t \leq t_n(p, \tilde{p}, \epsilon)$ is robust to small initial distributional mismatch due to its inherent stochasticity. For a more in depth analysis of the mathematics behind this intuitive idea we refer the reader to the work of Daras et al. [18].

**Sample dependent noise levels.** At a high level, our objective is to determine the $\epsilon$-merging time between the distribution of AlphaFold-predicted structures and that of experimentally resolved proteins. A key challenge arises from the fact that the AlphaFold distribution is highly heterogeneous in structural fidelity—that is, the accuracy with which AlphaFold predicts the true protein structure varies widely across samples. It is well established that short, structurally simple proteins are predicted with higher confidence, while longer and more complex proteins tend to yield lower-confidence predictions. This trend is illustrated in Figure 3B (Left). If we were to assign a single noise level across the entire AlphaFold dataset, we would need to select a relatively high noise level to accommodate the lowest-confidence predictions, particularly from long proteins. This would unnecessarily degrade the training signal for high-confidence structures—regardless of protein length—and limit the model's

ability to learn from clean supervision. To address this, we treat the AlphaFold dataset as a mixture of $K$ sub-distributions, $q_1, q_2, \ldots, q_K$, each representing a distinct confidence regime. We then assign each sub-distribution an appropriate noise level, sufficient to bring it $\epsilon$-close to the distribution of high-confidence structures under the same noise schedule. This formulation allows the model to effectively learn from high-confidence AlphaFold predictions and incorporate low-confidence structures in a controlled manner, mitigating the degradation typically caused by noisy training data.

A natural way to decompose the AlphaFold distribution into a mixture of quality-specific sub-distributions is to leverage AlphaFold's self-reported confidence metric—the average predicted Local Distance Difference Test (pLDDT) score—as a proxy for predicted structural fidelity. In particular, given a dataset $\mathcal{D} = \{(x_0^{(i)}, \text{pLDDT}^{(i)})\}_{i=1}^{N}$, we consider $K$ distributions (where $K$ is a hyperparameter to be chosen) with empirical observations for the $j$-th distribution being all the samples $\{(x_0^{(i)}, \text{pLDDT}^{(i)}) : c_{\min}^{(j)} \leq \text{pLDDT}^{(i)} \leq c_{\max}^{(j)}\}$, for some hyperparameters $c_{\min}^{(j)}, c_{\max}^{(j)}$.

**Choice of sub-distribution boundaries.** In this work, we adopt a deliberately simple and conservative strategy by partitioning the AlphaFold dataset into three discrete quality regimes based on the average pLDDT score: high-quality proteins (pLDDT $> 90$), medium-quality proteins (pLDDT in $[80, 90]$) and low-quality proteins (pLDDT in $[70, 80]$). We acknowledge that this discretization is coarse and that more principled alternatives may yield further improvements—for instance, by optimizing the bin boundaries or learning a continuous mapping from pLDDT to diffusion time. Despite the simplicity of our choices, our experimental results demonstrate that even a naive quality-aware decomposition can lead to important gains in performance across both short and long proteins. It is important to emphasize that there are two sources of benefit over filtering methods: 1) low-quality data (previously discarded) increases diversity, and 2) the distinction we do between medium-quality and high-quality data boosts designability. The population of proteins that have pLDDT lower than 70 has a very high merging time, and we did not see significant benefits from including them. To improve training efficiency and save computational resources, we decided to discard this subpopulation. We underline that training algorithms that are adaptive to local corruption (rather than global, i.e. average pLDDT) might be able to benefit from such proteins – we leave this direction for future work.

## 3.2 Ambient Protein Diffusion Algorithm

Our algorithm takes as input a dataset of protein structures together with their average pLDDT score, $\mathcal{D} = \{(x_0^{(i)}, \text{pLDDT}^{(i)})\}_{i=1}^{N}$, a diffusion schedule, $\sigma(t)$, and a mapping function $f : [0, 100] \mapsto \mathbb{R}^+$ that translates the average pLDDT value of a protein to its estimated $\epsilon$-merging time.

**Annotation stage.** The first step of the algorithm replaces each protein in the dataset with a noisy version of itself, where the noise level is determined by the mapping function $f$. This function is a hyperparameter for our algorithm – in our experiments, we opt for a rather simple choice (see Appendix Table 16 for a full description of our training configuration), but in principle this can be an arbitrary function defined by the user based on their specific domain knowledge or experimental findings. After this transformation, each protein can be treated as a sample from the target distribution convolved with a Gaussian at its assigned noise level. This transformation step is only performed once during dataset preprocessing, i.e., we replace the low-quality protein with a noisy version of itself *before* we start the training. This is important because adding different noise realizations across epochs can lead to recovery of the original low-quality protein if the noise is averaged out.

**Loss function.** After the annotation stage, we need to solve a training problem where we have data corrupted at different noise levels with additive Gaussian noise, as in [14, 16]. Hence, we can use the objective of Equation 2. Instead of directly applying the loss, we first need to rescale each time $t$ to account for the vanishing gradient effect that is due to the multiplicative factor $a(t)$. Specifically, we need to rescale the loss at time $t$ with: $w(t) = \frac{1}{a^2(t)} = \frac{\sigma^4(t)}{(\sigma^2(t) - \sigma^2(t_i))^2}$ such that we balance the different timesteps. This weighting is derived following the methodology of the EDM [30] paper (Appendix, p. 26, Section B6). We underline that this rescaling was not mentioned in the original paper of Daras et al. [16, 14], for training with noisy data. Yet, we find this rescaling critical for the success of our method. We hypothesize that the authors of [14, 16] did not encounter this issue because there were at most two noise levels considered, while in AF predicted protein structures there is a whole spectrum of assigned noise levels based on the predicted quality (measured by average

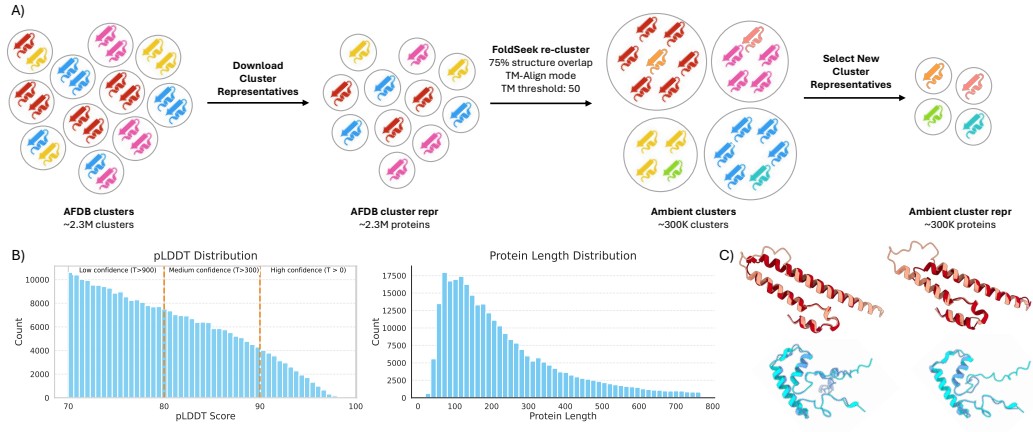

Figure 3: **Reclustering the AFDB cluster dataset to improve generative protein modeling.** (A) Starting from the 2.3M AFDB clustered dataset, we cluster the representatives with FoldSeek optimized for geometric similarity: alignment-type set to TM-Align, TM threshold set to 0.5, and coverage set to 0.75. This results in a 300K clusters (pLDDT > 70) from which we keep the representatives for training. (B) pLDDT and protein length statistics for our new training set. (C) Overlay of two Ambient clusters: Top row: Representative (beige; UniProt A0A2W1EPG1) overlaid with members A0A820X4G2 (left, red) and A0A446YZW1 (right, red). Bottom row: Representative (cyan; UniProt A0A395XDB6) overlaid with members A0A3B0YRI6 (left, blue) and L0S475 (right, blue). In the original AFDB cluster dataset, each of these six proteins was designated as the representative of its own cluster despite their similarities in structural features.

pLDDT) of a protein structure. We provide further details about the loss implementation in the Appendix (Section E) and pseudocode in Algorithm 1.

**Uniform Protein Sampling in terms of diffusion times.** To perform a training update for a diffusion model, we typically sample a point from the training distribution and then we uniformly sample the noise level $t$. However, since in our case we are dealing with noisy data, not all times $t$ are allowed for a given protein, i.e. a protein with $\text{pLDDT}^{(i)}$ is only used for times $t \geq f(\text{pLDDT}^{(i)})$. To avoid spending most of the training updates on very noisy proteins, we opt for sampling first the diffusion time and then select from the eligible proteins that can be used in that diffusion time. This strategy ensures balanced coverage across the diffusion trajectory—from low to high noise—while still leveraging the diversity of low-confidence structures (pLDDT < 80) in our training dataset.

**Summarizing:** Our algorithm requires three simple changes to the regular diffusion training: 1) an annotation stage (before training) where each low-quality protein is replaced with a noisy version of itself, 2) a change in the way we fetch samples from the dataset so that we do not overallocate training updates to highly noisy proteins and 3) a change in the loss function to account for the fact that for some proteins we do not have access to an uncorrupted structures.

## 3.3 Reclustering AFDB clusters for generative modeling applications

On top of our algorithmic contributions, we also reconsider the choices made for the training dataset. The AFDB clustered dataset [5] has been used to train several generative protein models [34, 23]. However, the original intent behind the clustering was to study structure evolution. Thus, the hyperparameters were chosen to obtain clusters of homologous structures, and the authors report that 97.4% of pairwise comparisons within clusters are conserved at the H-group (Homology) level of the ECOD hierarchical domain classification (median TM-score 0.71). While these FoldSeek hyperparameters are well-suited for evolutionary analysis, we found that the AFDB cluster dataset has a significant degree of structural duplication and near-duplication between clusters that are more distantly evolutionarily related (see Figure 3C). This structural redundancy leads to an imbalanced training set, where structural motifs from the larger protein superfamilies are overrepresented.

Given this finding, we hypothesize that the datasets for generative modeling of protein structures—particularly for backbone-based models— benefit more from clusters defined purely by geometric similarity. To that end, we construct a new clustering dataset derived from the AFDB cluster representatives, with an exclusive focus on structural topology. Specifically, these are the changes we made to the FoldSeek hyperparameters: we switch the alignment-type from 3Di+AA to TM-Align to improve fidelity, we use a TM-score threshold of 0.5, and we relax the alignment coverage from 0.9 to 0.75. We did the latter to improve clustering of AlphaFolded proteins with extended, unfolded N- or C-terminal regions (i.e., noodle tails) (Figure 3C). This approach produced a more balanced dataset that samples structural folds more uniformly, independent of their evolutionary relationships. Ablations that disentangle the contribution of this reclustering from our ambient training approach are given in Figure 4.

## 4 Experimental Results

We build on the Genie2 codebase [34]. Our model architecture follows the Genie2 architecture except that it is scaled larger, using 8 triangle layers as opposed to 5. We train Ambient Protein Diffusion in 3 stages with increasingly longer proteins, eventually reaching proteins up to length 768. For details on the training process, see Appendix Section E.3, and for details on metric,s see Appendix Section D. We underline that the computational cost of training our model is relatively low compared to the prior state of the art Proteína model. This is due to the decreased size of our model ($< 17M$ vs 200M) and training set ($\sim 290K$ vs $\sim 780K$). We further note that our goal is to develop models that perform well across a range of tasks, including long-protein generation, motif scaffolding, and more. To this end, we train only two models for the purposes of this paper: one model optimized for long-protein generation (Figure 1) and another optimized for short-protein generation (Figure 5).

### 4.1 Comparisons on unconditional generation of longer proteins

In Figure 1, we compare Ambient Protein Diffusion performance on generating backbone for proteins with length ranging from 300 to 800 residues. To directly compare with Proteína on long-protein generation, we adopt its three-stage training and evaluation protocol. During training, the maximum sequence length is capped at 768 residues. For evaluation, we sample 100 protein backbones at each target length and evaluate them using the designability and diversity metrics. Since Ambient Protein Diffusion builds on Genie2, we use the same sampling procedure—running 1000 diffusion steps with a noise scale of $\gamma = 0.6$. This noise scale parameter controls the trade-off between the designability and diversity by reducing the amount of stochasticity added in the reverse process, as it is typically done in the protein generative modeling literature (see Appendix E.4.1 for details).

Ambient Protein Diffusion achieves designability and diversity scores exceeding 90% for proteins between 300 and 500 residues, and maintains scores above 85% for lengths up to 700 residues. For 800-residue proteins, both metrics decline to 68%. Compared to Proteína, Ambient Protein Diffusion outperforms by 26% in designability and 91% in diversity at length 700, and by 24% and 45%, respectively, at length 800. At every protein length, Ambient Protein Diffusion's diversity is equal to its designability, indicating that every designable protein is unique. This is not the case for Proteína, where diversity scores consistently fall below designability, regardless of protein length.

Taken together, these results demonstrate the impact of ambient diffusion on backbone-based generative models and highlight the strength of Genie2's equivariant architecture. Our 17M parameter model trained on approximately 290K AlphaFold structures significantly outperforms a 200M-parameter transformer model trained on roughly 780K proteins. Our results show that smaller, more efficient models can surpass larger transformer baselines in both structural diversity and designability.

### 4.2 Ablating the significance of Ambient Diffusion

In comparison to Genie2, the starting point of our implementation, we made the following changes: 1) made the model bigger, 2) trained on longer proteins, 3) reclustered the dataset to optimize for geometric similarity rather than evolutionary similarity, and 4) used low pLDDT AlphaFold proteins as noisy data using our Ambient Protein Diffusion framework. To quantify how much of the improvement comes from the latter step, we train an Improved Genie2 without the Ambient Framework for training with corrupted data and we report results in Figure 4. We find that while

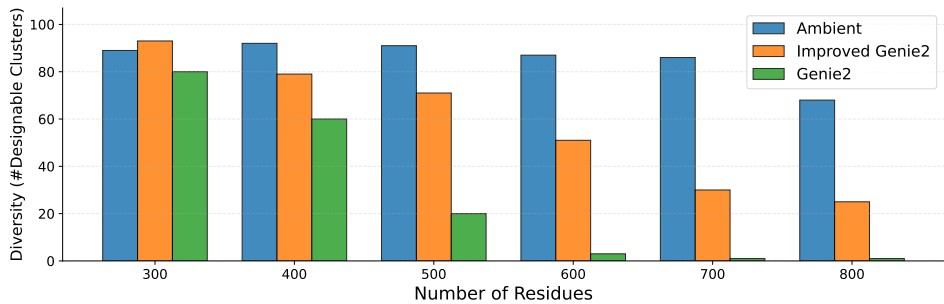

Figure 4: **Ablation to quantify the effect of the Ambient Protein Diffusion framework.** We improve upon Genie2 by making the architecture bigger, reclustering the dataset as in Section 3.3 and by finetuning on longer proteins (up to 768 aminoacids). The resulting improved Genie2, shown in Orange, outperforms Genie2 but still lags behind our Ambient Protein Diffusion Model, shown in blue. The only difference between the two models is that Ambient Protein Diffusion uses low pLDDT AlphaFold structures as noisy data, as explained in Section 3.2. Ambient Protein Diffusion consistently outperforms both the improved Genie2 baseline and Genie2, with increasingly significant improvements as sequence length grows.

our two models perform similarly on proteins of 300 residues, the designability and diversity of the baseline diminish on longer proteins. For proteins with 800 residues, the number of designable clusters drops from 68% to 25%. Ambient Protein Diffusion shows a marked improvement, maintaining a stable number of designable clusters. We underline that the difference comes solely from the training algorithm since the architecture, model size, hyperparameters, and inference algorithm remain the same. We note that this experiment only ablates the impact of treating AlphaFold data as noisy, and fixes the reclustered dataset (Section 3.3) for both models. We examine the impact of reclustering separately in Section B.3 and find that reclustering increases diversity; however, without Ambient training, the designability remains low. Both innovations are needed to achieve optimal results.

## 4.3 Comparisons on unconditional generation of shorter proteins

In this experiment, we evaluate the model on the unconditional generation of shorter proteins in Figure 5. We provide training details for the model optimized for short protein generation in the Appendix Section E.3.2.

Following the Genie2 protocol, we generate 5 structures for each sequence length from 50 to 256 residues, yielding a total of 1,035 structures. The generated structures are evaluated for both designability and diversity. In line with prior work, we sweep the noise scale $\gamma$ to explore the trade-off between designability and diversity. Ambient Protein Diffusion outperforms previous methods on both metrics, establishing a new Pareto frontier that achieves superior performance compared to all existing models, including Proteina.

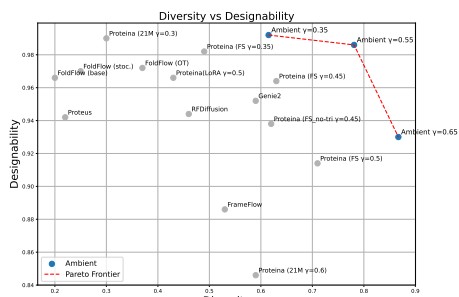

Figure 5: **Designability - diversity trade-off for short protein generation** (up to 256 residues). Ambient dominates completely the Pareto frontier between designability and diversity, while using a 12.88× smaller model. We further do so without using any higher-order sampler or (auto-) guidance method.

## 4.4 Novelty scores

The next step is to show that our method does not just memorize proteins, but instead it can generate novel designable structures that are distinct from the training set. Here, we compute the TM-novelty metric following the protocol of Geffner et al. [23]. However, we found two key issues when reproducing the literature TM-novelty scores, which we discuss in detail in Appendix Section D.3. First, since Spring 2025 FoldSeek resolved a bug in computing the alntmscore (Github issue 312), which makes all previous reported values incorrect. Second, we found that Geffner et al. [23]

Table 1: **PDB and AFDB TM-Novelty for short protein generation.** For each model, we sample 5 structures for each sequence length from 50 to 256 residues, yielding a total of 1,035 structures. A lower TM-Novelty score is better.

| Model | PDB Novelty ($\downarrow$) | AFDB Novelty ($\downarrow$) |
|---|---|---|
| *Ambient Proteins* | | |
| $\quad \gamma = 0.35$ | 0.774 | **0.848** |
| $\quad \gamma = 0.55$ | **0.773** | 0.851 |
| $\quad \gamma = 0.65$ | 0.774 | 0.858 |
| *Baselines* | | |
| Genie2 | 0.789 | 0.862 |
| RFDiffusion | 0.853 | 0.923 |
| Proteus | 0.842 | 0.884 |
| Chroma | 0.824 | 0.885 |
| Proteina (FS $\gamma = 0.35$) | 0.853 | 0.910 |
| Proteina (FS $\gamma = 0.45$) | 0.837 | 0.898 |
| Proteina (FS $\gamma = 0.5$) | 0.831 | 0.893 |
| Proteina (FS_no-tri $\gamma = 0.45$) | 0.832 | 0.891 |
| Proteina (21M $\gamma = 0.3$) | 0.889 | 0.932 |
| Proteina (21M $\gamma = 0.6$) | 0.854 | 0.905 |

inadvertently used the alnTM-Score from the row with the highest qTM-Score, rather than from the row with the highest alnTM-Score. To ensure accurate benchmarking with the literature, we recalculated TM-Novelty with the patched FoldSeek v10 for several baselines, explicitly selecting the max alnTM-Score for each query. We report results in Table 1. For backward compatibility, we also reproduced literature results using the unpatched FoldSeek v9 and the default qTM-Score sorting (Table 15). Moving forward, we strongly recommend that adopting FoldSeek v10 and always using the max alnTM-Score value to determine the novelty per query when computing TM-Novelty.

Using both versions of FoldSeek, Ambient Protein Diffusion sets new state-of-the-art TM-novelty scores on both the PDB and AFDB (588K) benchmarks. In the short-evaluation regime ($\leq 256$ AAs) using FoldSeek v10, we exceed the next-best model (Genie2) by 2.0% on PDB and 1.6% on AFDB—despite Genie2's restriction to proteins no longer than 256 AAs. Against Proteina, we further boost TM-novelty by 7.0% on PDB and 4.8% on AFDB. In the long-evaluation regime (300–800 AAs), we focus on comparison with Proteina. Here, we achieve TM-novelty scores of 0.682 on PDB and 0.740 on AFDB, representing improvements of 18.4% and 16.2%, respectively. Together, these results demonstrate that Ambient Protein Diffusion, driven by the Ambient loss and cluster dataset, produces the most novel proteins across both short and long-sequence settings.

## 4.5  Motif Scaffolding

As a final evaluation, we compare our method to prior work in motif scaffolding. The full results are shown in Appendix Figure 7 and in Appendix Tables 11 and 12. With $\gamma = 0.45$, Ambient Protein Diffusion generates 1,923 unique successful scaffolds for single-motif tasks, a significant improvement over Genie2's 1,445 [34] and performs comparably to a Proteína model (2,094 [23]), which is much larger (200M parameters vs 17M parameters) and is optimized specifically for motif scaffolding. For multi-motif scaffolding, Ambient Protein Diffusion generates 89 unique successful structures across 5 of the 6 problems, outperforming Genie2, which produces 40 and solves 4.

## 5  Conclusion

We introduced *Ambient Protein Diffusion*, a framework for protein structure generation that leverages low-confidence AlphaFold structures as a source of noisy training data. Ambient Protein Diffusion enables the generation of long protein structures with unprecedented levels of designability, diversity and novelty. Diversity increases as it can use low-confidence Alphafold structures that are typically discarded and designability increases as we separate the pristine quality proteins structures from the medium quality AlphaFold predictions. Ambient Protein Diffusion represents a foundational step toward robust de novo protein design at more natural, biologically relevant lengths.

# 6  Acknowledgements

This research has been supported by NSF Awards CCF-1901292, ONR grants N00014-25-1-2116, N00014-25-1-2296, a Simons Investigator Award, and the Simons Collaboration on the Theory of Algorithmic Fairness. The experiments were run on the Vista GPU Cluster through the Center for Generative AI (CGAI) and the Texas Advanced Computing Center (TACC) at UT Austin.

**Note on authorship.** William Daspit and Qiang Liu joined the team after an initial version of this project was submitted to NeurIPS. Both authors made major contributions to the final version. Due to the NeurIPS policy preventing authorship changes after submission, their names could not appear on the paper or the conference website. However, all the authors of this manuscript agree that they made very significant contributions and that this project would not be the same without their work. A version of this paper with the updated authorlist is available on bioarXiv: https://www.biorxiv.org/content/10.1101/2025.07.03.663105.

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

## A  Limitations and Future Work

This work represents a first step toward protein generative models that make better use of the synthetic structures from the AlphaFold Database. Nevertheless, there are several clear avenues for improvement. (i) Structure-quality metric. We rely on AlphaFold's self-reported pLDDT score—a coarse, residue-averaged confidence measure that can itself be noisy or misleading, (ii) Data coverage. We only use one representative per AFDB cluster rather than incorporating all available cluster members, and we build on the existing AFDB clustering rather than reclustering the full 214M–structure dataset, (iii) pLDDT–merging-time mapping. Our choice of how to translate pLDDT values into merging thresholds was driven by empirical tuning rather than by a systematic ablation study or principled selection criterion, (iv) Experimental validation. Ultimately, the real test of any generative model is whether its predictions hold up in the laboratory. We have yet to confirm our structures experimentally.

## B  Ablations

### B.1  Mapping function ablations

The algorithm described in Section 3.2 assumes a mapping function that maps a protein's pLDDT to a diffusion time after which the protein can be used for learning. For all the experiments of the paper, we used a discrete map (detailed in Table 16); in particular proteins that have pLDDT $\geq 90$ are used everywhere, proteins with pLDDT in $[80, 90)$ are used for only the last 400 diffusion steps (out of the 1000 total) and proteins that have pLDDT in $[70, 80]$ are only used for the last 100 diffusion steps. This mapping was chosen to represent three coarse categories; clean samples, semi-clean proteins and very noisy AlphaFold predictions. In this section, we ablate the choice of the mapping function to 1) show the robustness of our method to reasonable choices, and 2) show that it is possible to achieve improved results by optimizing this mapping.

We show results in Table 2 for small changes to the default choice used in the paper. Specifically, we change what happens when we inflate or deflate all the pLDDT boundaries in the discrete chosen mapping by 5 pLDDT points and we observe the differences in designability and diversity. For all these results, we use noise scale $\gamma = 0.6$ during sampling. As shown, our method is relatively robust, as small perturbations to the mapping don't lead to a very substantial change. In fact, the inflated thresholds lead to a model that belongs to the Pareto frontier of Figure 4 in the paper. Running the same model with $\gamma = 0.65$ leads to another Pareto point, achieving an (86.6, 0.882) designability-diversity pair.

| plddt_to_timestep | Designability | Diversity | Notes |
|---|---|---|---|
| (80,0) | 95.2 | 0.590 | Genie2 baseline |
| (90,0),(80,300),(70,900) | 98.6 | 0.781 | Paper choice |
| (85,0),(75,300),(70,900) | 96.4 | 0.780 | Deflated thresholds by 5 |
| (95,0),(85,300),(75,900) | 96.8 | 0.783 | Inflated thresholds by 5 |

Table 2: **Effect of changing the pLDDT-to-timestep function.** Slight adjustments to the thresholds lead to reasonable results, indicating that the method is robust.

Now that we established the robustness of our approach, we present further results that can be obtained by optimizing the pLDDT to timestep mapping. Towards that goal, we try using more discrete bins and we also explore two continuous generalizations to interpolate between the pLDDT boundaries used in the paper. In particular, we try a piecewise continuous function and a sigmoid function. To explain with an example, before all proteins with pLDDT in $(80, 90]$ were used for times 300-1000. Under the piecewise continuous generalization, a protein with pLDDT $x$ in $(80, 90]$ will be used for times $t >= 300 \cdot (90 - x)/(90 - 80)$, e.g. a protein with pLDDT 82 is used for times $t >= 240$. The results are given in Table 3.

As shown, by optimizing the mapping function, we can even outperform the results we obtained in the paper. That said, even a simplistic choice, as the ones we opted for in the paper, can already yield significant boosts over the Genie2 baseline.

| pLDDT to timestep mapping | Designability | Diversity | Pareto | Notes |
|---|---|---|---|---|
| (90,0),(80,300),(70,900) | 98.6 | 0.781 | ✓ Pareto optimal | Paper choice |
| Piecewise linear | 96.2 | 0.827 | ✓ Pareto optimal | |
| Sigmoid | 98.2 | 0.752 | | |
| (90,0),(85,300),(80,600),(75,900),(70,950) | 95.9 | 0.742 | | Extra bins 1 |
| (90,0),(85,200),(80,300),(75,850),(70,900) | 94.2 | 0.858 | ✓ Pareto optimal | Extra bins 2 |

Table 3: **Effect of different pLDDT-to-timestep mappings.** Several mappings achieve Pareto optimality, balancing designability and diversity.

## B.2 Weighting function ablation

In the paper, we used the weighting term $w(t) = \frac{1}{a^2(t)} = \frac{\sigma^4(t)}{(\sigma^2(t) - \sigma^2(t_i))^2}$ in the Ambient loss to balance the contribution of different diffusion times and avoid vanishing gradients. We ablate this choice in Table 4. As shown, removing that weighting leads to significant deterioration of the obtained performance, both in designability and diversity. The calculations for this weighting follow the EDM [30] methodology, Appendix p. 26, Section B6.

| Weighting | Designability | Diversity | Pareto |
|---|---|---|---|
| Paper choice | 98.6 | 0.781 | ✓ Pareto optimal |
| Constant (w=1) | 97.4 | 0.733 | |

Table 4: **Effect of weighting term $w(t) = \frac{\sigma^4(t)}{(\sigma^2(t) - \sigma^2(t_i))^2}$ in the loss.**

## B.3 Dataset reclustering ablation

In Figure 4 of the main paper, we fix the dataset to the geometric reclustered version we created in this work, and we ablate the effect of Ambient training. In this section, we perform additional experiments to understand the standalone value of geometric reclustering. We report results with all combinations of Ambient vs. no Ambient and reclustering vs. no reclustering in Table 5. As shown in the Table, reclustering leads to improved diversity, but without Ambient designability is low. Both innovations are needed to achieve the optimal performance.

| Setting | Dataset | Designability ↑ | Diversity ↑ |
|---|---|---|---|
| No Ambient | Genie2 | 95.2 | 0.590 |
| No Ambient | Re-clustered | 81.4 | 0.902 |
| Ambient | Genie2 | 96.4 | 0.501 |
| Ambient | Re-clustered | 98.6 | 0.781 |

Table 5: **Effect of dataset reclustering.** Re-clustering increases diversity, but without Ambient training, the designability is low. The combination of the two innovations yields the optimal results.

## B.4 Loss ablation

For low pLDDT structures, $x_0$ cannot be trusted. Instead, our loss involves predicting a noisy version of the low pLDDT protein, which can be trusted more than the original protein because, intuitively, the noise has erased some of the AlphaFold prediction mistakes. For completeness and to support our argument, we provide one further ablation where we train a model with the same setup as the short-protein generation in the paper, but using $x_0$ prediction loss instead of the proposed loss. Results are shown in Table 6.

For a more detailed discussion on the benefits of the Ambient loss over the original diffusion loss, we refer the reader to the work of Shah et al. [43].

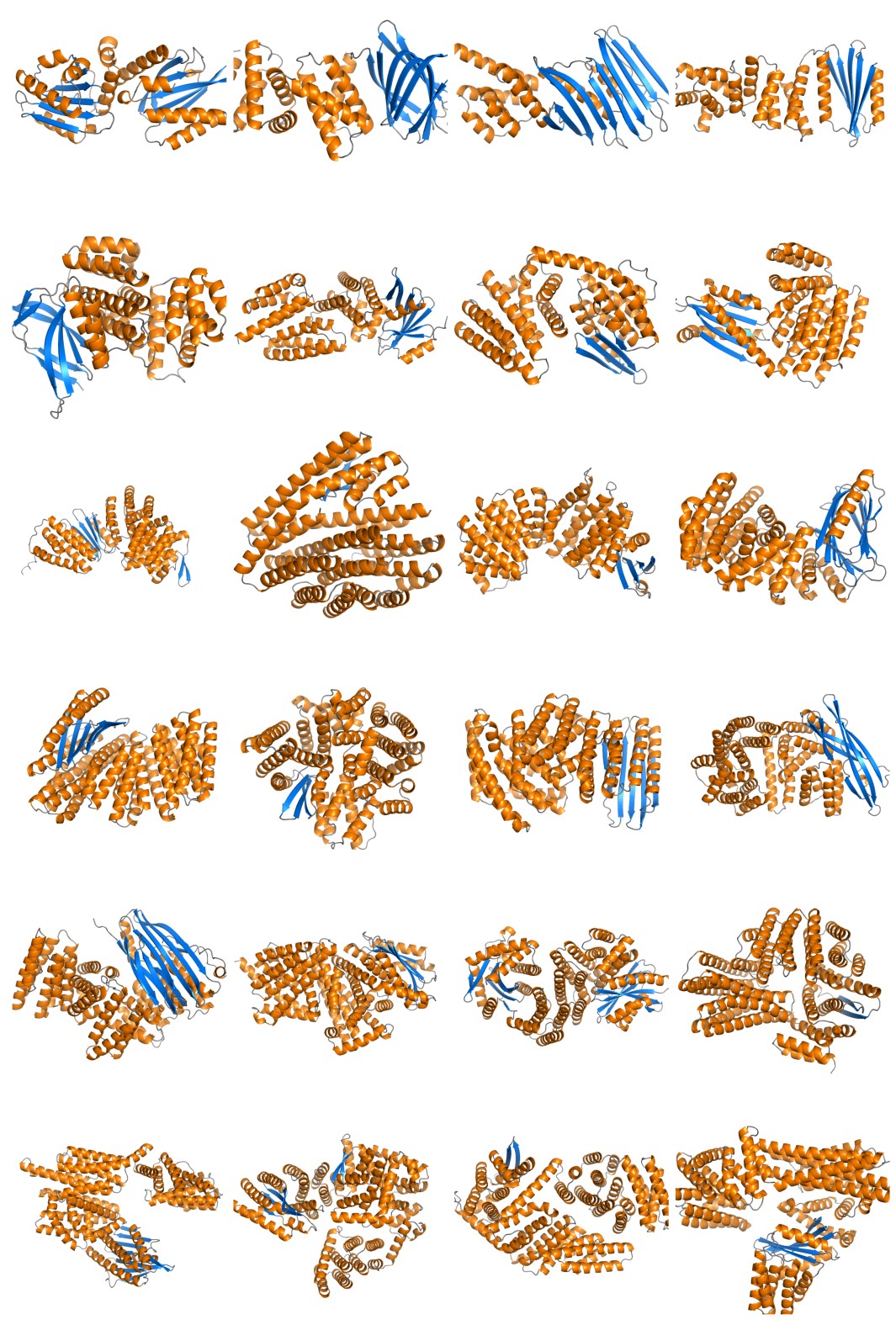

Figure 6: **Additional qualitative visualizations of unconditional generations.**

| Loss Type | Designability | Diversity |
|---|---|---|
| Ambient loss | 98.6 | 0.781 |
| $x_0$ prediction loss | 98.5 | 0.753 |

Table 6: **Effect of Ambient loss.** Ambient loss achieves slightly higher designability and diversity compared to $x_0$ prediction loss.

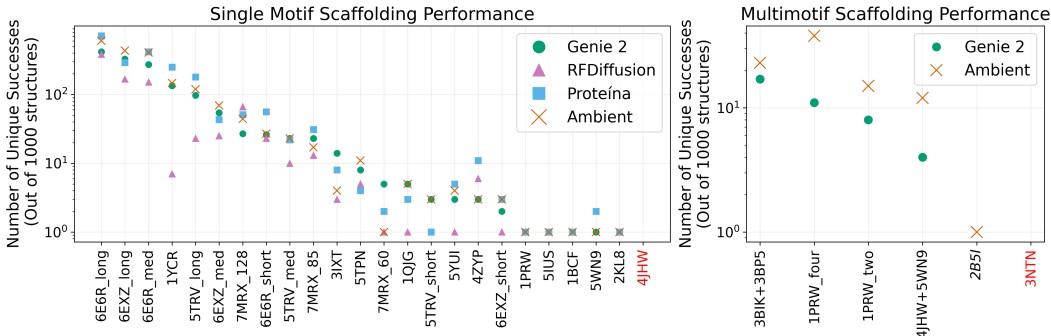

Figure 7: **Performance on Motif Scaffolding Tasks.** We compare Ambient Protein Diffusion to state-of-the-art models for motif scaffolding. The graphs show the number of unique successful scaffolds generated for each single- and multi-motif task. No model produced successful scaffolds for 4JHW and 3NTN. Only Ambient Protein Diffusion produced a valid solution for multi-motif scaffolding of *2B5I*.

## C  Additional Results

### C.1  Motif Scaffolding Results

We additionally compare our method to prior work on motif scaffolding in Figure 7, with full results provided in the supplement. Our evaluation follows the Genie2 benchmark, which comprises 24 single-motif and 6 multi-motif design tasks [34, 50]. For each task, we generate 1,000 scaffold samples using a noise scale of $\gamma = 0.45$. A design is considered successful if it (1) satisfies Genie2's motif designability criteria and (2) preserves the motif with an RMSD below 1Å. Among successful designs, a scaffold is counts as unique if its TM-score is at most 0.6 when compared to any other successful scaffold. A task is considered solved if at least one successful scaffold is generated.

With $\gamma = 0.45$, Ambient Protein Diffusion generates 1,923 unique successful scaffolds for single-motif tasks, a significant improvement over Genie2's 1,445 [34] and performs comparably to a Proteína model (2,094 [23]) that is much larger (200M parameters vs 17M parameters) and is optimized specifically for motif scaffolding. Notably, all methods solve a similar number of motifs – RFDiffusion solves 22 of the 24 tasks, while Ambient Protein Diffusion, Genie2, and Proteína each solve the same 23 tasks. For multi-motif scaffolding, Ambient Protein Diffusion generates 89 unique successful structures across 5 of the 6 benchmark problems, outperforming Genie2, which produces 40 and solves 4. Ambient Protein Diffusion performs particularly well on the 1PRW_four motif (38 vs. 11 successful structures) in which a scaffold is generated surrounding a calcium binding motif [48]. Overall, Ambient Protein Diffusion outperforms existing methods such as Genie2 and RFDiffusion on single-motif tasks and matches the performance of a Proteína model optimized specifically for motif-scaffolding.

We want to underline that when counting successes in motif scaffolding it is important to fix the scaffold length for a given motif for all the baselines. In particular, we noticed that increasing the protein length leads to consistent improvements. For fair comparison between models, we provide results for Ambient Protein Diffusion, Genie, and RFDiffusion, where increasing the scaffold lengths by 1.5 increased the total number of successes. Assuming we use the midpoint of the mininum and maximum scaffold lengths as provided in the Genie 2 benchmark (structures are provided by the RFDiffusion benchmark, but the lengths are from Genie), Genie generates 1327 successes, Ambient

Protein Diffusion generates 1849, and RFDiffusion generates 694. If we increase the length by 1.5, Genie, Ambient Protein Diffusion and RFDiffusion generate 2486, 3776 and 1033 successful scaffolds respectively.

## C.2 Secondary Structure Evaluations

In this section, we systematically evaluate the Secondary Structure Proportions of our generated proteins. Specifically, we first use biotite to assign secondary structure labels to all the generated proteins. Then, we look at the designable PDB files, we calculate the percentages of alpha/beta/loops, and then we average those over all designable PDBs. The results are shown in Table 7.

| Method | $\alpha$-Helix (%) $\downarrow$ | $\beta$-Sheet (%) $\uparrow$ |
|---|---|---|
| Genie2 | 72.7 | 4.8 |
| Proteina MFS 0.35 | 71.6 | 5.8 |
| Proteina MFS 0.45 | 68.1 | 6.9 |
| Proteina MFS 0.50 | 67.0 | 7.2 |
| Ambient Protein Diffusion (scale 0.5) | 69.9 | 7.9 |
| Ambient Protein Diffusion (scale 0.4) | 66.6 | 9.5 |

Table 7: **Secondary structure composition across methods.** Lower $\alpha$-Helix and higher $\beta$-Sheet percentages indicate more natural outputs.

As shown, Ambient Protein Diffusion generates proteins with more beta sheets and loops compared to Proteina and Genie2, illustrating progress towards designing more natural proteins. That said, all the models, including ours, show a bias towards alpha helices. One way to mitigate this bias is through secondary structure conditioning. In our code, we further release a model that can be conditioned on a per-residue SS label. As we increase the Classifier Free Guidance strength towards beta sheets, this model generates proteins with more interesting SS structure, as summarized in Table 8. Preliminary designability/diversity evaluations show weaker performance of this model compared to the unconditional models of our main paper. Optimally balancing structural diversity and strong designability/diversity metrics remains a challenging open problem.

| Guidance strength | $\alpha$-Helix (%) $\downarrow$ | $\beta$-Sheet (%) $\uparrow$ |
|---|---|---|
| 0.2 | 82.1 | 1.5 |
| 0.4 | 61.8 | 11.4 |
| 0.6 | 30.3 | 33.6 |
| 0.7 | 21.7 | 36.1 |

Table 8: **Effect of guidance strength on secondary structure composition.** Increasing the strength reduces $\alpha$-Helix content and increases $\beta$-Sheet content, indicating a shift towards more natural outputs.

## C.3 Complete Tabular Results

This section presents the full numerical tables corresponding to result figures shown in the text. Specifically,

- Table 9 enumerates the results in Figure 1 and Figure 4.
- Table 10 enumerate the results in Figure 5.
- Table 11 enumerates partial results in Figure 7.
- Table 12 enumerates partial results in Figure 7.

Table 9: **Long protein generation performance.** Best values per residue length are highlighted in bold. Ambient Proteins results are shown for $\gamma = 0.6$. Found in main text in Figure 1.

| Residue Length | Model | Designability (%) | Diversity (Clusters) |
|---|---|---|---|
| 300 | Ambient | 90 | 89 |
| | Proteina | **93** | 55 |
| | Proteus | **93** | 23 |
| | Improved Genie2 | **93** | **93** |
| | Genie2 | 81 | 80 |
| 400 | Ambient | **92** | **92** |
| | Proteina | 85 | 60 |
| | Proteus | 83 | 35 |
| | Improved Genie2 | 79 | 79 |
| | Genie2 | 60 | 60 |
| 500 | Ambient | **91** | **91** |
| | Proteina | 82 | 65 |
| | Proteus | 69 | 30 |
| | Improved Genie2 | 71 | 71 |
| | Genie2 | 20 | 20 |
| 600 | Ambient | **87** | **87** |
| | Proteina | 81 | 55 |
| | Proteus | 67 | 20 |
| | Improved Genie2 | 51 | 51 |
| | Genie2 | 2 | 3 |
| 700 | Ambient | **86** | **86** |
| | Proteina | 68 | 45 |
| | Proteus | 47 | 10 |
| | Improved Genie2 | 30 | 30 |
| | Genie2 | 1 | 0 |
| 800 | Ambient | **68** | **68** |
| | Proteina | 55 | 47 |
| | Proteus | 17 | 5 |
| | Improved Genie2 | 25 | 25 |
| | Genie2 | 0 | 0 |

Table 10: **Designability-diversity trade-off for short protein generation.** Designability and diversity for short protein generation. Found in main text in Figure 5.

| Model | Designability (%↑) | Diversity (↑) |
|---|---|---|
| *Ambient Proteins* | | |
| $\gamma = 0.35$ | **99.2** | 0.615 |
| $\gamma = 0.55$ | 98.6 | 0.781 |
| $\gamma = 0.65$ | 93.0 | **0.867** |
| *Baselines* | | |
| Genie2 | 95.2 | 0.59 |
| FoldFlow (base) | 96.6 | 0.20 |
| FoldFlow (stoc.) | 97.0 | 0.25 |
| FoldFlow (OT) | 97.2 | 0.37 |
| FrameFlow | 88.6 | 0.53 |
| RFDiffusion | 94.4 | 0.46 |
| Proteus | 94.2 | 0.22 |
| Proteina (FS $\gamma = 0.35$) | 98.2 | 0.49 |
| Proteina (FS $\gamma = 0.45$) | 96.4 | 0.63 |
| Proteina (FS $\gamma = 0.5$) | 91.4 | 0.71 |
| Proteina (FS_no-tri $\gamma = 0.45$) | 93.8 | 0.62 |
| Proteina (21M $\gamma = 0.3$) | 99.0 | 0.30 |
| Proteina (21M $\gamma = 0.6$) | 84.6 | 0.59 |
| Proteina (LoRA $\gamma = 0.5$) | 96.6 | 0.43 |

Table 11: **Performance on Single Motif Scaffolding Tasks** Ambient Protein Diffusion achieves superior results to Genie 2 and RFDiffusion and performs on par with Proteina. Crucially, our model achieves these results zero-shot, i.e., unlike Proteina, it is not optimized for motif scaffolding and still achieves comparable performance while being an order of magnitude smaller. Found in text in Figure 7.

| Motif Name | Genie 2 | RFDiffusion | Proteína | Ambient Protein Diffusion |
|---|---|---|---|---|
| **6E6R_long** | 415 | 381 | **713** | 601 |
| **6EXZ_long** | 326 | 167 | 290 | **432** |
| **6E6R_med** | 272 | 151 | **417** | 406 |
| **1YCR** | 134 | 7 | **249** | 146 |
| **5TRV_long** | 97 | 23 | **179** | 119 |
| **6EXZ_med** | 54 | 25 | 43 | **69** |
| **7MRX_128** | 27 | 66 | **51** | 44 |
| **6E6R_short** | 26 | 23 | **56** | 27 |
| **5TRV_med** | **23** | 10 | 22 | **23** |
| **7MRX_85** | 23 | 13 | **31** | 17 |
| **3IXT** | **14** | 3 | 8 | 4 |
| **5TPN** | 8 | 5 | 4 | **11** |
| **7MRX_60** | **5** | 1 | 2 | 1 |
| **1QJG** | **5** | 1 | 3 | **5** |
| **5TRV_short** | **3** | 1 | 1 | **3** |
| **5YUI** | 3 | 1 | **5** | 4 |
| **4ZYP** | 3 | 6 | **11** | 3 |
| **6EXZ_short** | 2 | 1 | **3** | **3** |
| **1PRW** | **1** | **1** | **1** | **1** |
| **5IUS** | **1** | **1** | **1** | **1** |
| **1BCF** | **1** | **1** | 1 | **1** |
| **5WN9** | 1 | 0 | **2** | 1 |
| **2KL8** | **1** | **1** | 1 | 1 |
| **4JHW** | 0 | 0 | 0 | 0 |
| **Total** | 1445 | 889 | **2094** | 1923 |

Table 12: **Performance on Multi Motif Scaffolding Tasks** Ambient Protein Diffusion achieves consistently superior results to the predecessor Genie-2 model, despite using the same architecture, i.e. the benefit comes from better use of the data. The motif 2B5I is only solved by Ambient Protein Diffusion. Found in text in Figure 7.

| Motif Name | Genie 2 | Ambient Protein Diffusion |
|---|---|---|
| **3BIK+3BP5** | 17 | **23** |
| **1PRW_four** | 11 | **38** |
| **1PRW_two** | 8 | **15** |
| **4JHW+5WN9** | 4 | **12** |
| **2B5I** | 0 | **1** |
| **3NTN** | 0 | 0 |
| **Total** | 40 | **89** |

Table 13: **Performance Comparison on Single Motif Scaffolding Tasks with Constant Midpoint Length**

| Motif Name | Genie2 | Ambient | RFDiffusion | Lengths |
|---|---|---|---|---|
| 6E6R_long | 406 | **601** | 208 | 108 |
| 6EXZ_long | 296 | **432** | 158 | 110 |
| 6E6R_med | 286 | **406** | 70 | 78 |
| 1YCR | 78 | **95** | 3 | 70 |
| 5TRV_long | 85 | **119** | 40 | 116 |
| 6EXZ_med | 46 | **69** | 17 | 80 |
| 7MRX_128 | 28 | **44** | 52 | 128 |
| 6E6R_short | 25 | **27** | 5 | 48 |
| 5TRV_med | **33** | 23 | 9 | 86 |
| 7MRX_85 | **24** | 17 | 12 | 85 |
| 3IXT | **5** | 1 | 1 | 62 |
| 5TPN | **2** | 1 | 1 | 62 |
| 7MRX_60 | **1** | **1** | **1** | 60 |
| 1QJG | 2 | 2 | **103** | 78 |
| 5TRV_short | 2 | 1 | **4** | 56 |
| 5YUI | 1 | **2** | 1 | 75 |
| 4ZYP | 1 | **2** | **2** | 40 |
| 6EXZ_short | 1 | 1 | **2** | 50 |
| 1PRW | **1** | **1** | **1** | 82 |
| 5IUS | **1** | **1** | **1** | 99 |
| 1BCF | **1** | **1** | **1** | 124 |
| 5WN9 | **1** | **1** | **1** | 42 |
| 2KL8 | **1** | **1** | **1** | 79 |
| 4JHW | **0** | **0** | **0** | 75 |
| Total | 1327 | **1849** | 694 | |

# D    Evaluation Metrics

Evaluation of a protein generative model is challenging and there have been a few metrics that have been proposed. In what follows, we explain standard metrics in the protein-generative modeling literature that we will use in our Experimental Results section. Our experiments report using Proteína's definitions of the metrics when possible.

## D.1    Designability

Designability (also referred to as refoldability) assesses the structural plausibility of generated proteins. Given a generated backbone, ProteinMPNN [19] generates eight plausible amino acid sequences for that backbone. ESMFold then folds each sequence and the resulting eight structures are compared to the original backbone. The self-consistency RMSD (scRMSD) is defined as the smallest root mean squared deviation between the generated backbone and each of the eight refolded structures. A backbone is considered *designable* if scRMSD < 2 Å and designability is defined as the percentage of generated backbones that meet this criterion.

## D.2    Diversity

Diversity quantifies the structural variability among the generated proteins. Designable backbones are clustered using Foldseek with a TM-score threshold of 0.5. Diversity is then defined as:

$$\text{Diversity} = \frac{\text{Number of Designable Clusters}}{\text{Number of Designable Samples}}.$$

This metric reflects the proportion of structurally distinct (i.e., non-redundant) designable backbones among all designable samples.

Table 14: **Performance Comparison on Single Motif Scaffolding Tasks with 1.5x Length**

| Motif Name | Genie2 | Ambient | RFDiffusion | Lengths |
|---|---|---|---|---|
| **6E6R_long** | 439 | **720** | 243 | 162 |
| **6EXZ_long** | 526 | **848** | 193 | 165 |
| **6E6R_med** | 393 | **633** | 74 | 117 |
| **1YCR** | 321 | **459** | 25 | 105 |
| **5TRV_long** | 64 | **102** | 14 | 174 |
| **6EXZ_med** | 349 | **490** | 19 | 120 |
| **7MRX_128** | 14 | **23** | **84** | 192 |
| **6E6R_short** | 145 | **215** | 6 | 72 |
| **5TRV_med** | 82 | **109** | 5 | 129 |
| **7MRX_85** | 23 | **42** | 15 | 127 |
| **3IXT** | **48** | 27 | 1 | 93 |
| **5TPN** | 16 | **40** | 13 | 93 |
| **7MRX_60** | 9 | **10** | 2 | 90 |
| **1QJG** | 0 | 2 | **323** | 117 |
| **5TRV_short** | **11** | 10 | 2 | 84 |
| **5YUI** | 4 | **9** | 1 | 112 |
| **4ZYP** | 6 | **13** | 6 | 60 |
| **6EXZ_short** | **31** | 17 | 2 | 75 |
| **1PRW** | **1** | **1** | **1** | 123 |
| **5IUS** | **1** | **1** | **1** | 149 |
| **1BCF** | 1 | **2** | 1 | 186 |
| **5WN9** | 1 | **2** | 0 | 63 |
| **2KL8** | **1** | **1** | **1** | 118 |
| **4JHW** | 0 | 0 | **1** | 112 |
| **Total** | 2486 | **3776** | 1033 | |

Table 15: **PDB and AFDB TM-Novelty for short protein generation using FoldSeek-v9 and max qTM-score row.** We recompute all values using Geffner et al. [23] method. Numbers reported by Geffner et al. [23] are shown in parenthesis. These numbers are reported for backwards comparisons only and we strongly encourage the community to use the corrected TM-Novelty scores reported in the main text (see Appendix D.3).

| Model | PDB Novelty ($\downarrow$) | AFDB Novelty ($\downarrow$) |
|---|---|---|
| *Ambient Proteins* | | |
| $\gamma = 0.35$ | **0.604** | **0.663** |
| $\gamma = 0.55$ | 0.606 | 0.671 |
| $\gamma = 0.65$ | 0.608 | 0.673 |
| *Baselines* | | |
| Genie2 | 0.621 (0.63) | 0.685 (0.69) |
| RFDiffusion | 0.711 (0.71) | 0.779 (0.77) |
| Proteus | 0.741 (0.74) | 0.766 (0.76) |
| Chroma | 0.686 (0.69) | 0.732 (0.74) |
| Proteina (FS $\gamma = 0.35$) | 0.727 (0.71) | 0.783 (0.77) |
| Proteina (FS $\gamma = 0.45$) | 0.709 (0.69) | 0.769 (0.75) |
| Proteina (FS $\gamma = 0.5$) | 0.698 (0.69) | 0.760 (0.75) |
| Proteina (FS_no-tri $\gamma = 0.45$) | 0.702 (0.69) | 0.759 (0.76) |
| Proteina (21M $\gamma = 0.3$) | 0.811 (0.81) | 0.841 (0.84) |
| Proteina (21M $\gamma = 0.6$) | 0.761 (0.72) | 0.797 (0.77) |

### D.3  Novelty

**Metric definition.** Novelty is a metric that assesses the uniqueness of the generated backbones in comparison to existing structures in a database. We compute the novelty score with respect to both AFDB and PDB datasets following Geffner et al. [23]. To compute novelty, we measure the structural similarity of each designable protein to those in the dataset using FoldSeek's easy-search command used by Proteina:

```
foldseek easy-search <path_sample> <database_path> <out_file> <tmp_path>
-alignment-type 1 -exhaustive-search -tmscore-threshold 0.0 -max-seqs
10000000000 -format-output query,target,alntmscore,lddt
```

For each designable backbone, we keep the max alntmscore value rather than the alntmscore value of the first row, which is the max qtmscore value. The novelty of the dataset is the average of these maximum alntmscore values, representing how distinct our generated structures are from the proteins in the reference database (i.e., we do `df.groupby("query")["alntmscore"].max().mean()`). Perhaps counterintuitively, high novelty is not desired since it implies high similarity to the existing database.

**Bug in novelty computation.** Several of our evaluation metrics —TM-Diversity and TM-Novelty— depend on FoldSeek's TM-score implementation. In Fall 2024, however, FoldSeek developers identified a bug in the `alntmscore` output (see Github issue 312 titled "alntmscore output is wrong" for details), which means that all previously reported TM-based metrics in the literature that did not use FoldSeek v10 (release 10-941cd33) are incorrect. Additionally, we found that Geffner et al. [23] mistakenly computed TM-novelty by taking the alnTM-Score from the row with the highest qTM-Score, rather than from the row with the highest alnTM-Score. This oversight arises because Foldseek's easy-search command, by default, sorts its output in descending order by qTM-Score—irrespective of the requested output format.

To ensure accurate and comparable benchmarking with the literature, we recalculated TM-Novelty with the patched FoldSeek v10 (release 10-941cd33), explicitly selecting the maximum alnTM-Score for each query. For backward compatibility, we also reproduced literature results using the unpatched FoldSeek v9 (release 9-427df8a) and the default max qTM-Score row. Moving forward, we strongly recommend that the community adopt FoldSeek v10 and always sort using the alnTM-Score output to determine the maximum TM-Score per query and correctly compute TM-Novelty. Using both versions of FoldSeek, Ambient Protein Diffusion sets new state-of-the-art TM-novelty scores on both the PDB and AFDB (588K) benchmarks.

## E  Full Training Algorithm and Implementation Details

### E.1  Additional Implentation Details

**Loss buffer.** The loss rescaling introduced in the main paper ensures balanced weighting across noise levels. At the same time, it also introduces a potential instability: the loss explodes as $\sigma(t)$ approaches $\sigma(t_i)$. To mitigate this instability, we define a buffer zone around each protein's assigned noise level. Specifically, given a protein's assigned noise level $t_i$, it is only used during training at timesteps $t + \tau$, where $\tau$ is a buffer hyperparameter that controls the exclusion margin. This constraint prevents the model from encountering degenerate gradient behavior near the rescaling boundaries and is only applied to medium and low confidence structures (pLDDT < 90). We underline that is similar to how in normal diffusion there is a buffer time zone around $t = 0$ that is never sampled.

**Ambient in high-noise regime.** As explained in the main paper, each protein is only used for a subset of diffusion times according to its average pLDDT value. The proteins that have super high PLDDT ($> 90$) are considered clean data and can be used with the normal training objective. However, as found in [43], using the Ambient training objective for high-noise might be useful even if clean data is available. Intuitively, this objective prevents memorization and promotes diversity in the outputs. We ablated this design choice, and we found a slight increase in diversity for the same designability by using this. Hence, we used this tool from [43] for all our Ambient Protein Diffusion trainings.

## E.2 Algorithm

We provide the full algorithm in Algorithm 1. We commit to open-sourcing our code and models to facilitate the broader adoption of our method from the community.

---

**Algorithm 1** Ambient Protein Diffusion: Training Algorithm.

---

**Require:** untrained network $h_\theta$, dataset $\mathcal{D} = \{(x_0^{(i)}, \text{pLDDT}^{(i)})\}_{i=1}^N$, pLDDT to diffusion time mapping function $f : [0, 100] \mapsto \mathbb{R}^+$, noise scheduling $\sigma(t)$, batch size $B$, diffusion time $T$, buffer $\tau$.

1: $\tilde{\mathcal{D}} \leftarrow \left\{ \left( x_0^{(i)} + f(\text{pLDDT}^{(i)})\epsilon^{(i)}, f(\text{pLDDT}^{(i)}) \right) | (x_0^{(i)}, \text{pLDDT}^{(i)}) \in \mathcal{D}, \epsilon^{(i)} \sim \mathcal{N}(0, I_d) \right\}$ ▷ Noise each point in the training set according to its pLDDT and get (noisy, noise level) pairs.

2: **while** not converged **do**

3:     $t_s^{(1)}, ..., t_s^{(B)} \leftarrow$ Sample uniformly B times in $[0, T]$ ▷ Sample diffusion times for this batch.

4:     $\tilde{\mathcal{D}}_p \leftarrow \text{shuffle}(\tilde{\mathcal{D}})$ ▷ Shuffle dataset.

5:     $\text{loss} \leftarrow 0$ ▷ Initialize loss.

6:     $\text{pos} \leftarrow 0$ ▷ Initialize index at shuffled dataset.

7:     **for** $i \in [1, B]$ **do**

8:         **while** True **do** ▷ find the first eligible point

9:             $y, t_y \leftarrow \tilde{\mathcal{D}}_p[\text{pos}]$

10:             **if** $t_y \geq t_s^i + \tau$ **then**

11:                 break

12:             **else**

13:                 $\text{pos} \leftarrow \text{pos} + 1$ ▷ Move to the next point in the dataset.

14:             **end if**

15:         **end while**

16:         $\epsilon \sim \mathcal{N}(0, I)$ ▷ Sample noise.

17:         $t \leftarrow t_s^{(i)}$ ▷ Time to be used in this training update.

18:         $t_i \leftarrow t_y$ ▷ Assigned time based on the PLDDT value

19:         $x_{t_i} \leftarrow y$ ▷ Noised point to the assigned time.

20:         $x_t \leftarrow x_{t_i} + \sqrt{\sigma^2(t) - \sigma^2(t_i)}\epsilon$ ▷ Add additional noise.

21:         $\alpha(t, t_i) \leftarrow \frac{\sigma^2(t) - \sigma^2(t_i)}{\sigma^2(t)}$.

22:         $w(t, t_i) \leftarrow \frac{\sigma^4(t)}{(\sigma^2(t) - \sigma^2(t_i))^2}$. ▷ Loss reweighting.

23:         $\text{loss} \leftarrow \text{loss} + w(t, t_i) \left\| \alpha(t, t_i) h_\theta(x_t, t) + (1 - \alpha(t, t_i))x_t - x_{t_i} \right\|^2$ ▷ Ambient loss

24:     **end for**

25:     $\text{loss} \leftarrow \frac{\text{loss}}{B}$ ▷ Compute average loss.

26:     $\theta \leftarrow \theta - \eta \nabla_\theta \text{loss}$ ▷ Update network parameters via backpropagation.

27: **end while**

---

## E.3 Model and Training Hyperparameters

### E.3.1 Hyperparameters for model optimized for long generation.

We train Ambient Protein Diffusion in 3 stages with increasingly longer proteins. In the first stage, we train on proteins from 50 to 256 residues for 200 epochs on our ambient clusters dataset using the representatives ($\sim$ 196,000 proteins). Since we increased the batch size to 384 items, we adopted a learning rate schedule to improve convergence [25]. We train with the AdamW optimizer with a maximal learning rate of $1.0 \times 10^{-4}$. During the second and third stage, we include additional cluster representatives of at most 512 and 712 residues, which scales our dataset to $\sim$269,000 and $\sim$291,000 proteins respectively. Training is performed on 48 GH200 GPUs and runs in 18, 48, and 48 hours for each stage respectively. We underline that the computational cost of training our model, while significant, is still relatively low compared to the Proteína's estimated 14 days training on 128 A100 GPUs. This is due to the decreased size of our model ($< 17$M vs 200M) and training set ($\sim$ 290K vs $\sim$ 780K).

Table 16 includes a more thorough list of the hyperparameters used for our experiments.

Table 16: **Hyperparameters of the diffusion protein model.** Dashes (-) indicate that the value is the same as the previous column. The Ambient walls correspond to the assigned diffusion times based on the protein's pLDDT (times are from 1 to 1000). Proteins with pLDDT $> 90$ are used everywhere. Proteins with pLDDT $> 80$ are used for times in $[600, 1000]$ and proteins with pLDDT $> 70$ are used for times in $[900, 1000]$. We underline that these hyperparameters were not particularly optimized, and even more benefits might be observed by properly tuning these values.

| Hyperparameter | Genie2 | *Ambient* (Stage 1) | Stage 2 | Stage 3 |
|---|---|---|---|---|
| **Diffusion** | | | | |
| Number of timesteps | 1,000 | - | - | - |
| Noise schedule | Cosine | - | - | - |
| Allowed times: | - | $\begin{cases} [1, 1000], \ \text{pLDDT} \geq 90 \\ [600, 1000], \ 90 > \text{pLDDT} \geq 80 \\ [900, 1000], \ 80 > \text{pLDDT} \geq 70 \end{cases}$ | - | - |
| **Model Architecture** | | | | |
| Single feature dimension | 384 | - | - | - |
| Pair feature dimension | 128 | - | - | - |
| Pair transform layers | 5 | 8 | 8 | 8 |
| Triangle dropout | 0.25 | - | - | - |
| Structure layers | 8 | - | - | - |
| **Training** | | | | |
| Optimizer | AdamW | - | - | - |
| Number of training proteins | 586k | 196k | 269k | 291k |
| Number epochs | 40 | 200 | 50 | 20 |
| Warmup iterations | 10,000 | 1,000 | 500 | 100 |
| Total batch size | 384 | 384 | 96 | 48 |
| Learning rate | $1.0 \times 10^{-4}$ | $1.0 \times 10^{-4}$ | $1.0 \times 10^{-5}$ | $1.0 \times 10^{-5}$ |
| Weight decay | 0.05 | - | - | - |
| Minimum protein length | 20 | 20 | 50 | 50 |
| Maximum protein length | 256 | 256 | 512 | 768 |
| Minimum mean pLDDT | 80 | 70 | 70 | 70 |
| **Compute Resources** | | | | |
| Number of GPUs | 48 | 48 | 48 | 48 |
| Training time | 18 hr | 18hr | 48hr | 48hr |

### E.3.2 Hyperparameters for model optimized for short generation.

The Ambient Protein Diffusion model used in this experiment was trained on a dataset filtered with a TM-Align threshold of 0.4 (as opposed to 0.5), resulting in a training set of approximately 90K cluster representative proteins. While it is well known that protein pairs with TM-scores above 0.5 typically share the same fold, and those below 0.5 generally do not, we find that the trade-off between designability and diversity is sensitive to the underlying structural heterogeneity of the dataset. Notably, clustering with a TM-align threshold of 0.4, which corresponds to less than a 1% chance of shared global topology, slightly outperforms the 0.5 threshold, which reflects a $\sim$38% probability of topological similarity [54].

### E.4 Sampling

#### E.4.1 Noise scale

In diffusion modeling, one designs a forward Ito corruption process:

$$\mathrm{d}X_t = f(X_t, t)\mathrm{d}t + g(t)\mathrm{d}B_t, \tag{4}$$

defined by the drift function $f(\cdot, \cdot)$ and the noise coefficient $g(\cdot)$. This process gets initialized at a distribution $p_0$ and diffuses over time, defining smoother densities $p_t$. Due to a remarkable result by Anderson [3], sampling from $p_0$ is achieved by running the reverse process:

$$\mathrm{d}X_{T-t} = (-f(X_{T-t}, T-t) + g^2(T-t)\nabla \log p_{T-t}(X_{T-t}))\mathrm{d}t + g(T-t)\mathrm{d}B_{T-t}, \tag{5}$$

initialized at $p_T$.

However, in the context of protein generative models, it has been observed that sampling from a discretized version of the reverse process of Equation 6 does not lead to good performance as measured by the available metrics. Hence, it is common practice in the protein generative modeling literature to sample from a tilted measure using the process:

$$\mathrm{d}X_{T-t} = (-f(X_{T-t}, T-t) + g^2(T-t)\nabla \log p_{T-t}(X_{T-t}))\mathrm{d}t + g(T-t)\sqrt{\gamma}\mathrm{d}B_{T-t}, \tag{6}$$

where the parameter $\gamma$ controls the stochasticity added to the generation. Typically, this parameter is set to values $\gamma < 1$ leading to more designable proteins at the expense of reduced diversity in the generated samples. The goal is often to optimally control the trade-off between designability and diversity, i.e. to be able to produce a wide range of structurally and functionally diverse proteins. Unless stated otherwise, for the experiments in this paper, we use $\gamma = 0.6$ (as done in Genie2 [34]).

#### E.4.2 Hyperparameters and sampling methods

For sampling, we follow the exact same parameters as Genie2. In particular, we run 1000 sampling steps using a simple first-order discretization of Equation (6). We underline that results could be further enhanced by using more advanced sampling techniques such as autoguidance [31] (used in Proteina [23]), higher-order samplers [30] and test-time scaling [38] methods.

