# OpenReview forum: "Ambient Proteins - Training Diffusion Models on Noisy Structures"
_NeurIPS.cc/2025/Conference — NeurIPS 2025 spotlight_

### Official Review · Reviewer_Rk4E · 2025-07-01

**Clarity:** 3
**Significance:** 4
**Originality:** 3
**Rating:** 5
**Confidence:** 4

**Summary:**

The paper introduce a framework (”*Ambient Diffusion”*) for training diffusion models that accounts for noisy (low-confidence) structures - which describes a considerable fraction of the predicted structure databases (e.g., from AlphaFold and ESMFold).  Extends recent work on training diffusion models data with a known corruption process by extending it to domains where the corruption process is more complex and a priori unknown.  They consider errors in the predicted structure to be such a process and the pLDDT from AlphaFold to be a reasonable proxy the structural accuracy.

They build the method on the insight that regardless of how two distributions deviate; adding Gaussian noise causes the KL-divergence between them to contract.  Thus they define a “epsilon-merging time” as the time at which the added noise causes two distributions to have D_KL < epsilon.  They bin structures into low (70≤pLDDT≤80), medium (80≤pLDDT≤90) and high (90≤pLDDT) confidence, and at the start of training apply Gaussian noise at the level is calculated to be the noise at the epsilon-merging time.  Intuitively, this means that the samples have the corruption incorrect predictions “erased” by applying sufficient noise that this error is not longer perceptible.  It follows that one can only train on samples at noise levels above this threshold (so practically, e.g., at low noise you only train on high confidence structures).  It should also be noted that, separate from the *Ambient Diffusion* framework, they also re-cluster the AFDB data aiming for improved “geometric diversity”.

Experimentally, they validate the approach by applying it to a Genie-like model architecture and training, and compare to a suite of structure generation models — most pertinently the current state-of-the-art Proteina.  On unconditional generation (300-800 residues) Ambient Protein outperforms all baseline models by large margin on standard designabilty and diversity metrics, (despite using fewer parameters and less training compute than Proteina).  Unconditional generation of shorter proteins (50-256 residues), they again demonstrate an improved better pareto front on these metrics when varying the temperature of the generative process.  Ablations show that the  *Ambient Diffusion* framework improves over an otherwise identical diffusion-like training.  Finally, they also consider the motif scaffolding benchmark from Genie2, showing reasonable performance - albeit not as clearly state-of-the-art as in the previous experiments.

**Questions:**

**Q1** How the mapping function, f, (line 203) is calculated in practice? Is it empirically calculated separately for each of the bins (low, medium, high confidence) and so not per-protein but per-bin?  What is the reference distribution against which these are compared when computing the epsilon-merging time?

**Q2** Can you help with my interpretation of the ablations in Figure 5.

**a** As I understand, Ambient vs Baseline is a direct ablation of the proposed framework, showing it has a significant impact on diversity.  Can you present the same results on designabilty also?

**b** Is it possible to infer the impact of the re-clustering of AFDB in isolation from any of the presented results? While Genie2 presumably doesn’t use the re-clustered data, it’s also a smaller model so can’t be taken as a clean ablation.

In the event that mis-interpretations or oversights in my review are highlighted during the review process, I of course may update my score - though overall I am already a fan of the paper and will be recommending an acceptance.  With suitable clarification on the few points above I would be willing to increase scores for Quality and/or Clarity.

**Ethical Concerns:**

["NO or VERY MINOR ethics concerns only"]

**Final Justification:**

For the reasons laid out in my review and subsequent disscusion; I keep my original positive recommendation.

**Limitations:**

Yes

**Quality:**

3

**Strengths And Weaknesses:**

**Strengths**

- The proposed idea is elegant, well-motivated and leads to clear practical performance gains.
- The experimental evaluation is compelling: suitable metrics and models are chosen and ablations are provided.
- The paper is very well written, providing good depth of discussion and intuition for several non-obvious technical steps.

**Weaknesses**

- While possibly my misunderstanding, one point where the method description is less clear is relating to the pre-processing of the samples prior to training.  Specifically, it’s not clear to me exactly how the mapping function, f, (line 203) is calculated and what the reference distribution is.  See Q1.
- The significance of the proposed re-clustering of AFDB could be better discussed.  Specifically, I am not sure this can be inferred from the ablations presented, see Q2.

---

> ### Author Rebuttal · Authors · 2025-07-31
>
> We are very pleased to see the thoughtful and very positive feedback of the Reviewer! In what follows, we do our best to address the Reviewer's questions and potential concerns. In particular:
>
> * We present further improved motif scaffolding results.
> * We clarify the ablation of Figure 5 and present additional ablations regarding reclustering.
> * We better explain our current choice of the mapping fuction and we provide ablations.
>
> ### Mapping function
>
> **Reviewer**: "it’s not clear to me exactly how the mapping function is calculated. Is it empirically calculated separately for each of the bins and so not per-protein but per-bin? What is the reference distribution?"
>
> Excellent question and apologies for not having it clear in the first place. The reference distribution is the distribution of "real proteins", i.e. proteins observed in nature. In the paper, we approximate this with the set of proteins for which AlphaFold is super confident (average pLDDT > 90).
>
> Ideally, each protein would be mapped to each time separately. The mapping should rely on some proxy for the quality of the protein (e.g. pLDDT). In our paper, we did not use a continuous mapping -- instead we discretized proteins into bins based on their average pLDDT and each bin got an annotation time.
>
> Specifically, we used the function `f` defined by the `plddt_to_timestep` mapping:
>
> (90, 0), (80, 300), (70, 900)
>
> Proteins with:
> - **pLDDT > 90** are used throughout all steps (0 to 1000).
> - **pLDDT in (80, 90]** are used from step 300 onward.
> - **pLDDT in (70, 80]** are used only from step 900 onward.
>
> **We will now provide ablations on this choice of mapping**. We report (designability, diversity) pairs for the short-protein generation benchmark (Figure 4), as it is computationally infeasible to do long protein training for all these studies.
>
> First, we show what happens when we slightly change the function f we used in the paper in two opposite directions.
>
> | `plddt_to_timestep` | Designability | Diversity | Notes |
> |---------------------|---------------|-----------|-------|
> | (80,0) | 95.2 | 0.590 | Genie2 baseline |
> | **(90,0),(80,300),(70,900)** | **98.6** | 0.781 | Paper choice |
> | (85,0),(75,300),(70,900) | 96.4 | 0.780 | Deflated thresholds by 5 |
> | (95,0),(85,300),(75,900) | 96.8 | **0.783** | Inflated thresholds by 5 |
>
> As shown, our method is relatively robust, as small perturbations to the mapping don’t lead to a very substantial change. In fact, the inflated thresholds lead to a model that belongs to the Pareto frontier of Figure 4 in the paper. Running the same model with scale=0.65 leads to another Pareto point, achieving an (86.6, 0.882) designability-diversity pair.
>
> #### Continuous mapping functions and more bins
>
> We further present continuous generalizations. We now try:
>
> i) a piecewise linear function.
> ii) a sigmoid function.
> iii) more bins
>
> E.g. before all proteins with pLDDT in (80, 90] were used for times 300-1000. Under i), a protein with pLDDT x in (80, 90] will be used for times t>=300 * (90 - x) / (90 - 80), e.g. a protein with pLDDT 82 is used for times t>=240.
>
> | `plddt_to_timestep`| Designability | Diversity | Pareto         | Notes        |
> |-----------------------------|---------------|-----------|----------------|--------------|
> | (90,0),(80,300),(70,900)    | **98.6**      | 0.781     | ✅ Pareto optimal | Paper choice |
> | Piecewise linear            | 96.2          | 0.827     | ✅ Pareto optimal |              |
> | Sigmoid                     | 98.2          | 0.752     |                |              |
> | (90,0),(85,300),(80,600),(75,900),(70,950) | 95.9 | 0.742 |                | Extra bins 1 |
> | (90,0),(85,200),(80,300),(75,850),(70,900) | 94.2 | **0.858** | ✅ Pareto optimal | Extra bins 2 |
>
> By optimizing the mapping function, we can even outperform the results we obtained in the paper.
>
> ### Clarifications on Figure 5 and Data ablations
>
> **Reviewer**: "Can you help with my interpretation of the ablations in Figure 5."
>
> Apogologies for not having it as clear in the paper. Figure 5 compares **three models**:
>
> 1. **Original Genie2**
>    Trained on short proteins with the original Genie2 dataset (pLDDT > 80 filtering, no reclustering).
>
> 2. **Improved Genie2 (Baseline w/o Ambient)**
>    Trained on longer proteins using the **re-clustered data, filtered at pLDDT > 70**.
>
> 3. **Ambient Diffusion**
>    Trained on longer proteins using the **re-clustered data** filtered at pLDDT > 70,
>    *plus* the Ambient methodology for utilizing low-quality pLDDT samples.
>
> In summary, this figure **fixes the dataset** (re-clustered, pLDDT > 70) and **ablates the Ambient methodology**.
>
> **Reviewer**: "Can you present the same results on designabilty also?"
>
> We present Designability ad Diversity numbers for Original Genie2, Improved Genie2, Proteina and our Ambient Protein Diffusion below (shown as (designability, diversity)):
>
> | Length | Original Genie2 | Improved Genie2 | Proteina     | Ambient Protein Diffusion |
> |--------|------------------|------------------|--------------|----------------------------|
> | L300   | (81,90)          | (93,93)          | (93,55)      | (90,89)                    |
> | L400   | (60,60)          | (79,79)          | (85,60)      | (92,92)                    |
> | L500   | (20,20)          | (71,71)          | (82,65)      | (91,91)                    |
> | L600   | (2,3)            | (51,51)          | (81,55)      | (87,87)                    |
> | L700   | (1,0)            | (30,30)          | (68,45)      | (86,86)                    |
> | L800   | (0,0)            | (25,25)          | (55,47)      | (68,68)                    |
>
> As shown, **our Ambient Protein Diffusion model massively outperforms all baselines.**
>
> #### Additional novelty numbers
>
> We further report novelty numbers for the models of the paper (in short and long generation) to truly show that our model can generate unique proteins:
>
> | Setting                   | Model                    | PDB Novelty ↓ | AFDB Novelty ↓ |
> |---------------------------|--------------------------|----------------|-----------------|
> | Long Protein Generation   | Ambient Protein Diffusion | **0.682**      | **0.740**       |
> |                           | Proteina                  | 0.836          | 0.883           |
> | Short Protein Generation  | Ambient Protein Diffusion | **0.774**      | **0.848**       |
> |                           | Proteina                  | 0.832          | 0.891           |
>
> ### Data Ablations
>
> **Reviewer**: "is it possible to infer the impact of the re-clustering of AFDB in isolation?"
>
> We agree with the Reviewer that we need to separately ablate the need for having a re-clustered dataset. Results below:
>
> | Setting                     | Dataset              | Designability ↑ | Diversity ↑ |
> |----------------------------|----------------------|------------------|--------------|
> | No Ambient                 | Genie2      | **95.2**             | 0.590        |
> | No Ambient                 | Re-clustered         | 81.4             | **0.902**        |
> | Ambient                    | Genie2      | 96.4             | 0.501        |
> | Ambient                    | Re-clustered         | **98.6**         | **0.781**    |
>
> As shown, **reclustering leads to improved diversity**.
>
> ### (Bonus): Improved motif scaffolding results
>
> In their summary, the Reviewer mentions: "[the authors] consider the motif scaffolding benchmark from Genie2, showing reasonable performance - albeit not as clearly state-of-the-art"
>
> We investigated this and we noticed that **the scaffold length is not fixed for our baselines, and this length impacts the number of unique successful structures.**
>
> In general, for a given motif, the models produce higher numbers of unique successes for longer lengths. In the paper, the numbers reported for Ambient are using the same lengths as Genie 2. We generated samples using 1.5x the average length Genie 2 uses for each motif, and **the number of sucesses icreases from approx. 2K to approx 4K (almost doubles!).**
>
> **Regarding inconsistencies:** For 5TPN, RFDiffusion sets the length to be between 39 and 99 residues, while Genie 2 generates samples between 50 and 75 residues. For 1QJG, Proteína sets the length range to be 95-152, while Genie 2 sets it to 53-103. The lengths used for the other motifs are not provided for either RFDiffusion or Proteína. We cannot fairly compare the performance without sampling the same lengths.
>
> Results for 1.5x legths are shown below:
>
> | Motif      | Genie2 Length | 1.5× Length |
> |------------|------|------|
> | 6E6R_long  | 510  | 720  |
> | 6EXZ_long  | 417  | 848  |
> | 6E6R_med   | 220  | 633  |
> | 1YCR       | 113  | 459  |
> | 5TRV_long  | 73   | 102  |
> | 6EXZ_med   | 51   | 490  |
> | 7MRX_128   | 33   | 23   |
> | 6E6R_short | 16   | 215  |
> | 5TRV_med   | 27   | 109  |
> | 7MRX_85    | 10   | 42   |
> | 3IXT       | 3    | 27   |
> | 5TPN       | 6    | 40   |
> | 7MRX_60    | 1    | 10   |
> | 1QJG       | 2    | 2    |
> | 5TRV_short | 1    | 10   |
> | 5YUI       | 0    | 9    |
> | 4ZYP       | 4    | 13   |
> | 6EXZ_short | 1    | 17   |
> | 1PRW       | 1    | 1    |
> | 5IUS       | 1    | 1    |
> | 1BCF       | 1    | 2    |
> | 5WN9       | 1    | 2    |
> | 2KL8       | 1    | 1    |
> | 4JHW       | 0    | 0    |
> | **Total**  |1923  |3776  |
>
> **We believe that our rebuttal strongly addresses the Reviewer's questios**. If so, we would be very grateful if the Reviewer considers further reinforcing their support for our work. We remain at the Reviewer's availability if there are any remaining concerns.

---

> > ### Comment · Reviewer_Rk4E · 2025-08-05
> >
> > Thank you for the detailed and considered response, which addresses the open questions I had.  One additional point that comes up from reviewing your comments is with respect to the Data Ablations.
> >
> > From the table it appears that the ambient setting does not materially improve performance when paired with the original Genie 2 dataset, and that the data reclustering is having a significant impact.  To be clear, this is not a critique over the method - I retain that it is an interesting contribution materially impacting training performance with a simple (meant in a good way) trick.  However, I believe that if the final version of the paper was explicit on the mapping function and data ablation (i.e., includes these results in a SM where not already there, and references the key takeaways in the main text) it would ensure it was transparent and helpful to future practitioners.
> >
> > With all of that said, I keep my original positive recommendation.

---

> > > ### Author Response · Authors · 2025-08-06
> > > **Thank you**
> > >
> > > Thank you for reading our rebuttal and for all your time and insightful feedback throughout the Review process.
> > > Regarding the data ablations, it seems that the reclustering improves diversity, but designability drops unless Ambient is used. The combination of the two contributions leads to diverse and designable models.
> > >
> > > We will most definitely include all the ablations we run throughout the rebuttal in the Supplementary Material, reference them in the main text and provide some key takeaways for practitioners. We are hoping that the community will adopt this idea as we agree with the Reviewer that it is an interesting but simple trick that leads to substantial performance benefits.
> > >
> > > We will further make all models and code public to make the adoption easier.

---

### Official Review · Reviewer_Yqh5 · 2025-07-02

**Clarity:** 2
**Significance:** 3
**Originality:** 4
**Rating:** 5
**Confidence:** 4

**Summary:**

Ambient Protein Diffusion (APD) tackles noisy AlphaFold2 backbones by treating each structure’s confidence score as a cue for how much Gaussian noise to inject before training; the ε-merging-time rule ensures that, after this adjustment, ambient diffusion can be applied to unknown corruption. The authors first re-cluster the AlphaFold Database by geometric similarity to create a 292 k-structure, fold-diverse dataset, then fine-tune a 17 M-parameter Genie-style network for less than five days on 48 GH200 GPUs. Despite its modest size and compute budget, the resulting model surpasses the much larger 200 M-parameter Proteína on backbone designability and diversity up to 700 amino acids, beats vanilla diffusion baselines on short-protein tasks, and produces more novel motif scaffolds. Overall, APD shows that embracing low-confidence structural predictions—not discarding them—can broaden protein-space coverage and deliver state-of-the-art generation efficiency.

**Questions:**

- Provide a short formal derivation (or appendix) of the ambient loss starting from the ϵ-merging definition.
- Add ablations for:
    1. alternative *f*(pLDDT) mappings (e.g., linear, sigmoid, more bins);
    2. re-clustering + standard training vs. original clustering + ambient training.
- Quantify secondary-structure composition—e.g., helix/strand/loop percentages—mirroring Genie 2’s Figure 2.

**Ethical Concerns:**

["NO or VERY MINOR ethics concerns only"]

**Final Justification:**

Given the strength of the theoretical derivation and the robustness of the noise mapping mechanism, I believe this paper holds significant value and should be accepted.

**Limitations:**

- Reliance on pLDDT as a noise oracle
The method equates AlphaFold2 confidence with corruption level; when pLDDT is poorly calibrated (e.g., false-high scores on mis-threaded regions or low scores on well-modelled flexible loops) the ε-merging schedule injects the wrong amount of noise, limiting fidelity for those motifs.
- Dataset-inherited AlphaFold bias
Training almost entirely on AF2 predictions means the generator learns AlphaFold’s statistical prior rather than experimental structure distributions, potentially reproducing AF2 hallucinations or over-confident helices/loops absent in PDB data.

**Paper Formatting Concerns:**

No Formatting Concerns

**Quality:**

3

**Strengths And Weaknesses:**

### **Strengths**

- **Methodological novelty** – The paper is the first to *generalise ambient diffusion to arbitrary, unknown corruption processes* by introducing the ϵ-merging-time concept and a per-sample noise schedule.
- **Low-quality data usage** – By mapping AF2 structure’s pLDDT to an adaptive diffusion start-time, the model **learns from low-confidence predictions instead of discarding them**, turning previously unusable data into a driver of diversity and designability gains.
- **Geometric re-clustering dataset** – Re-clustering 1.29 M AFDB representatives *for geometric rather than evolutionary similarity* yields a balanced 292 K-structure set that covers folds more uniformly; the procedure (TM-Align 0.5, 0.75 coverage, etc.) is spelt out for replication.
- **Strong performance** –A 16.7 M-parameter model achieves > 85 % designability *and* diversity up to 700 aa and still 68 % at 800 aa—beating Proteína by 26 / 91 pp at 700 aa and establishing a new Pareto frontier on ≤ 256 aa sequences.

### **Weaknesses**

1. **Loss derivation is mostly intuitive.**

    Although the paper frames “ϵ-merging time” as a principled way to handle unknown corruption, it never derives the final ambient-diffusion loss from first principles. A concise mathematical induction (or at least a sketch) would clarify why the chosen loss is sound.

2. **Underspecified mapping from pLDDT to noise.**

    The critical function *f*(pLDDT) and its three-bin implementation are hand-tuned; no ablation or sensitivity analysis justifies the choices. Without empirical evidence the method may over- or under-weight certain confidence ranges.

3. **No quantitative evidence for helix bias.**

    The authors acknowledge the model “still favours alpha-helical structures” but supply no numbers. Reproducing Genie 2’s Figure 2 (helix/strand ratios vs. natural proteins) would make the bias—and any progress—transparent.

4. **Missing dataset-vs-method ablation.**

    Two innovations are introduced simultaneously: (i) geometric re-clustering of AFDB for diversity and (ii) ambient training on noisy structures. The paper lacks an ablation table to separate their individual contributions.

---

> ### Author Rebuttal · Authors · 2025-07-31
>
> We are pleased that the Reviewer appreciated the novelty of our approach, the motivation, and the strong experimental performance of our method. The Reviewer has concerns regarding: i) the theoretical validity of our loss, ii) limited ablations about a) our mapping function and b) our dataset reclustering, and iii) missing evaluations on secondary structure. We provide clarifications and numerous ablations and evaluations to address these concerns.
>
> * We provide a derivation of our loss function from first principles.
> * We ablate the choice of the mapping function by providing:
>     * Robustness analysis to mispecification
>     * More bins
>     * Continuous generalizations
> * We ablate the dataset reclustering and the Ambient loss separately.
> * We show improvements over the prior state-of-the-art on secondary structure evaluation.
>
> ### Loss Derivation
>
> **Reviewer:**: "Provide a short formal derivation of the ambient loss [...]"
>
> Let's use $t_n$ to denote the $\epsilon$-merging time. Let also $p_0$, $q_0$ be the high-quality and low-quality distributios, respectively. Let's also use R.Vs $X_0, Y_0$ to denote samples from $p_0, q_0$. We know that $\mathrm{KL}(p_{t_n} || q_{t_n}) \leq \epsilon$. The goal is to learn the function $\mathbb E[X_0 | X_t=\cdot]$.
>
> If we had samples from $p_{t_n}$, then it is possible to learn the function of interest, $\mathbb E[X_0 | X_t=\cdot]$, without ever seeing a sample from $p_0$. This result has been proven in the paper "Consistent Diffusion Meets Tweedie" in the context of diffusion models and relates to the Noisier2Noise method from the Signal Processing community. The idea is that we can show the following identity for all times $t \geq t_n$:
>
> $\mathbb E[X_0 | X_t=x_t] = (1 - c_t) \mathbb E[X_{t_n} | X_t=x_t] + c_t x_t$ (for some $c_t$ that is a function of the noise level).
>
> (see Consistent Diffusion Meets Tweedie, p14, Eq. A28.)
>
> The proof of this identity is quite easy: we can apply Tweedie's formula twice for the R.V. $X_t$, one expressing it as a fuction of $X_0$ and one expressing it as a function of $X_{t_n}$. Using this identity and the definition of the conditional expectation as the best denoiser in l2 sense, we arrive at the objective we have in the paper.
>
> The only issue in all this discussion is that we don't have samples from $p_{t_n}$, we have samples from $q_{t_n}$. Hence, we can only learn $\mathbb E[Y_0 | Y_t = x_t]$ through $\mathbb E[Y_{t_n} | Y_t = x_t]$ for all times $t \geq t_n$. How far is $\mathbb E[Y_{t_n} | Y_t = x_t]$ from the desired $\mathbb E[X_{t_n} | X_t = x_t]$? The answer is not much, because of the $\epsilon$-merging. We can exactly bound this by using Girsanov's theorem that expresses the KL error as an error of the conditional expectations (For a reference, see the work Sampling is as easy as learning the score, page 11, Equation 5.4 of the arXiv) Hence, since the KL error is bounded, the same holds for the errors of the conditional expectations.
>
> We hope this proof sketch makes sense. We will add a detailed derivation in the Camera Ready version.
>
> ### Secondary Structure Evaluation
>
> **Reviewer**: "Quantify secondary-structure composition—e.g., helix/strand/loop percentages—mirroring Genie 2’s Figure 2."
>
> Following the Reviewer’s recommendation, we systematically evaluate the Secondary Structure Proportions of our generated proteins. Specifically, we first use biotite (as in Proteina) to assign secondary structure labels to all the generated proteins. Then, we look at the designable PDB files, we calculate the percentages of alpha/beta/loops, and then we average those over all designable PDBs. The results are shown below:
>
> | Method             | α ↓  | β ↑  |
> |--------------------|------|------|
> | Genie2             | 72.7 | 4.8  |
> | Proteina MFS scale 0.35           | 71.6 | 5.8  |
> | Proteina MFS scale 0.45           | 68.1 | 6.9  |
> | Proteina MFS scale 0.50           | 67.0 | 7.2  |
> | Ambient scale 0.5        | 69.9 | 7.9  |
> | Ambient scale 0.4        | 66.6 | **9.5** |
>
>
> As shown, Ambient Protein Diffusion generates proteins with more beta sheets and loops compared to Proteina and Genie2, illustrating progress towards designing more natural proteins.
>
> #### SS conditional model
> Over the past two months (since the submission), we further trained a model that can be conditioned on a per-residue SS label. As we increase the Classifier Free Guidance strength towards beta sheets, this model generates proteins with more interesting SS structure:
>
> | Strength Setting | α-Helix (%) ↓ | β-Sheet (%) ↑ |
> |------------------|---------------|----------------|
> | 0.2              | 82.1          | 1.5            |
> | 0.4              | 61.8          | 11.4           |
> | 0.6              | 30.3          | 33.6           |
> | 0.7              | 21.7          | **36.1**       |
>
> We plan to make these models available together with the code release upon acceptance of our work. We will further add all the Ablations and the Secondary Structure Evaluations in the paper.
>
>
>
> ### More Dataset & Ambient Ablations
>
> **Reviewer**:  "The paper lacks an ablation table to separate their individual contributions."
>
> Figure 5 in the paper directly ablates the effect of using Ambient Diffusion, as it shows the effect of having it and not having it on the same exact dataset (reclustered filtered at pLDDT > 70).
>
> We agree with the Reviewer that we need to separately ablate the need for having a re-clustered dataset. Results below:
>
> | Setting                     | Dataset              | Designability ↑ | Diversity ↑ |
> |----------------------------|----------------------|------------------|--------------|
> | No Ambient                 | Genie2      | **95.2**             | 0.590        |
> | No Ambient                 | Re-clustered         | 81.4             | **0.902**        |
> | Ambient                    | Genie2      | 96.4             | 0.501        |
> | Ambient                    | Re-clustered         | **98.6**         | **0.781**    |
>
> As shown, **reclustering leads to improved diversity**. Combining the reclustered dataset with Ambient gives the best results.
>
> ### Mapping function ablations
>
> **Reviewer**: "Add ablations for: alternative f(pLDDT) mappings."
>
>
> #### Robustness
>
> In the paper, we used the function `f` defined by the `plddt_to_timestep` mapping:
>
> (90, 0), (80, 300), (70, 900)
>
> Proteins with:
> - **pLDDT > 90** are used throughout all steps (0 to 1000).
> - **pLDDT in (80, 90]** are used from step 300 onward.
> - **pLDDT in (70, 80]** are used only from step 900 onward.
>
> We will now show what happens when we slightly change the function f we used in the paper in two opposite directions.
>
> | `plddt_to_timestep` | Designability | Diversity | Notes |
> |---------------------|---------------|-----------|-------|
> | (80,0) | 95.2 | 0.590 | Genie2 baseline |
> | **(90,0),(80,300),(70,900)** | **98.6** | 0.781 | Paper choice |
> | (85,0),(75,300),(70,900) | 96.4 | 0.780 | Deflated thresholds by 5 |
> | (95,0),(85,300),(75,900) | 96.8 | **0.783** | Inflated thresholds by 5 |
>
>
> As shown, our method is relatively robust, as small perturbations to the mapping don’t lead to a very substantial change. In fact, the inflated thresholds lead to a model that belongs to the Pareto frontier of Figure 4 in the paper. Running the same model with scale=0.65 leads to another Pareto point, achieving an (86.6, 0.882) designability-diversity pair.
>
> #### Continuous mapping functions and more bins
>
> We further present continuous generalizations of our original choice. In the paper, we chose a very rough categorization of the proteins into “high-quality”, medium-quality, and “low-quality” for didactic purposes (L191-200). We now try:
>
> i) a piecewise linear continuous generalization.
> ii) a sigmoid.
> iii) more bins
>
> E.g. before all proteins with pLDDT in (80, 90] were used for times 300-1000. Under i), a protein with pLDDT x in (80, 90] will be used for times t>=300 * (90 - x) / (90 - 80), e.g. a protein with pLDDT 82 is used for times t>=240.
>
> | `plddt_to_timestep`| Designability | Diversity | Pareto         | Notes        |
> |-----------------------------|---------------|-----------|----------------|--------------|
> | (90,0),(80,300),(70,900)    | **98.6**      | 0.781     | ✅ Pareto optimal | Paper choice |
> | Piecewise linear            | 96.2          | 0.827     | ✅ Pareto optimal |              |
> | Sigmoid                     | 98.2          | 0.752     |                |              |
> | (90,0),(85,300),(80,600),(75,900),(70,950) | 95.9 | 0.742 |                | Extra bins 1 |
> | (90,0),(85,200),(80,300),(75,850),(70,900) | 94.2 | **0.858** | ✅ Pareto optimal | Extra bins 2 |
>
> As shown, by optimizing the mapping function, we can even outperform the results we obtained in the paper. That said, even a simplistic choice, as the ones we opted for in the paper, can yield significant boosts.
>
> ### Limitations of pLDDT and AlphaFoldDB
>
> **Reviewer**: "pLDDT is (sometimes) poorly calibrated".
>
> This is true -- pLDDT is known to be unreliable sometimes. We will acknowledge this limitation more strongly in the paper. That said, our method can use any other metric of quality (other than pLDDT). For the purposes of the paper, we chose to use pLDDT because existing pipelines (Genie2/Proteina/RFDiffusion) were filtering proteins based on that score.
>
> **Reviewer**: "generator might learn AlphaFold’s statistical prior rather than experimental structure distributions"
>
> If the quality estimation oracle (in this case, pLDDT) is assumed to be perfect, then that's not going to happen because AlphaFold's bad predictions and hallucinations will be treated as nosiy data. In fact, that's the problem that previous methods are suffering from and our method fixes, at least partially. That said, pLDDT is not perfect (see above) and some biases might remain.
>
> **We believe that our rebuttal strongly addresses the Reviewer’s concerns, and we would be grateful if the Reviewer considers upgrading their score to reflect this.**

---

> > ### Comment · Reviewer_Yqh5 · 2025-08-05
> >
> > Thank you for the excellent work. Your explanation has resolved my previous confusion regarding the loss derivation. Including the detailed derivation will undoubtedly enhance the clarity and impact of your work. I found the evaluation on secondary structure particularly impressive—it clearly demonstrates that training the same model under different mechanisms can result in more diverse secondary structure generation. Additionally, the model’s robustness to varying noise mappings is a notable strength.
> >
> > That said, the use of pLDDT as an indicator for noise level remains a concern, as it is still a poorly calibrated metric. I hope future work explores more principled and elegant approaches to noise mapping.
> >
> > Overall, since most of my concerns have been thoroughly addressed, I will raise my score accordingly.

---

> > > ### Author Response · Authors · 2025-08-06
> > > **Thank you!**
> > >
> > > Thank you for your time and feedback -- it significantly helped us improve our work. We will make sure to add all the results from the rebuttal, including the analytic derivation of the loss function and the secondary structure evaluations, as promised.
> > >
> > > We are very pleased that most of the concerns have been addressed. We certainly agree that there are better ways of measuring the quality of a structure than pLDDT, and we will be exploring this direction in future works!

---

### Official Review · Reviewer_zoX9 · 2025-07-03

**Clarity:** 3
**Significance:** 3
**Originality:** 3
**Rating:** 5
**Confidence:** 2

**Summary:**

This paper presents Ambient Protein Diffusion, a diffusion-based method for protein structure generation that incorporates low-confidence AlphaFold2 predictions rather than discarding them. The approach treats structures with lower pLDDT scores as corrupted data and adjusts the diffusion process accordingly, using per-sample noise schedules to align noisy predictions with reliable structures. Experimental results show improvements in diversity and designability compared to baselines.

**Questions:**

- How does the performance change as the number or boundaries of pLDDT bins are varied, or if a continuous mapping is used? Is there a principled way to optimize binning or the $f$ function?
- Given the bias toward alpha-helical structures noted in the conclusion, have the authors examined generated structure topologies in more detail (e.g., secondary structure proportions, SCOP/CATH distribution)? What steps could mitigate this bias?

**Ethical Concerns:**

["NO or VERY MINOR ethics concerns only"]

**Final Justification:**

The concerns about hyperparameters' sensitivity and generation diversity are addressed.

**Limitations:**

yes

**Quality:**

3

**Strengths And Weaknesses:**

Strengths:
- The central innovation is a general framework for exploiting structures of unknown corruption by merging their distributions using a sample-adaptive noise schedule, thus integrating all available synthetic AF data without coarse filtering. This presents a flexible, data-efficient approach likely to be relevant for generative modeling in other domains with noisy scientific data.
- Extensive experiments demonstrate strong performance in protein generation and motif scaffolding tasks, with the method outperforming state-of-the-art models in diversity and designability across most conditions.


Weaknesses:
- While ablation versus standard diffusion is addressed, and clustering thresholds are discussed, there is insufficient exploration of the sensitivity/robustness of the reported gains to hyperparameters, such as choice or granularity of pLDDT bins, mapping function $f$, or diffusion schedule.
- Authors note that the method tends to generate alpha-helical structures preferentially , which remains an important unsolved limitation. This could limit the diversity of discovered proteins and undercuts one of the paper's main claims of achieving unprecedented diversity.

---

> ### Author Rebuttal · Authors · 2025-07-31
>
> We thank the Reviewer for their time and thoughtful feedback. The Reviewer appreciated the novelty of our method, our strong experimental results and the potential relevance of our approach for scientific domains with noisy data. The only concerns seem to be about i) missing ablation studies about the mapping function and ii) missing secondary structure evaluation.
>
> In what follows, we address these raised issues by:
> * providing numerous ablations about the mapping function
> * evaluating comprehensivel our secondary structure performance
> * Providing novelty numbers
> * Presenting further improved Secondary Structure results using a conditional model.
>
> For all our ablations on the mapping function, we report (designability, diversity) pairs for the short-protein generation benchmark (Figure 4 of the paper), as it is computationally infeasible to do long protein training for all these studies.
>
> ### Mapping function ablations
>
>
> #### Robustness
>
> **Reviewer:** "there is insufficient exploration of the sensitivity/robustness".
>
> In the paper, we used the function `f` defined by the `plddt_to_timestep` mapping:
>
> (90, 0), (80, 300), (70, 900)
>
> Proteins with:
> - **pLDDT > 90** are used throughout all steps (0 to 1000).
> - **pLDDT in (80, 90]** are used from step 300 onward.
> - **pLDDT in (70, 80]** are used only from step 900 onward.
>
> We will now show what happens when we slightly change the function f we used in the paper in two opposite directions.
>
> | `plddt_to_timestep` | Designability | Diversity | Notes |
> |---------------------|---------------|-----------|-------|
> | (80,0) | 95.2 | 0.590 | Genie2 baseline |
> | **(90,0),(80,300),(70,900)** | **98.6** | 0.781 | Paper choice |
> | (85,0),(75,300),(70,900) | 96.4 | 0.780 | Deflated thresholds by 5 |
> | (95,0),(85,300),(75,900) | 96.8 | **0.783** | Inflated thresholds by 5 |
>
>
> As shown, our method is relatively robust, as small perturbations to the mapping don’t lead to a very substantial change. In fact, the inflated thresholds lead to a model that belongs to the Pareto frontier of Figure 4 in the paper. Running the same model with scale=0.65 leads to another Pareto point, achieving an (86.6, 0.882) designability-diversity pair.
>
> #### Continuous mapping functions and more bins
>
> We further present continuous generalizations of our original choice. In the paper, we chose a very rough categorization of the proteins into “high-quality”, medium-quality, and “low-quality” for didactic purposes (L191-200). We now try:
>
> i) a piecewise linear continuous generalization.
> ii) a sigmoid.
> iii) more bins
>
> E.g. before all proteins with pLDDT in (80, 90] were used for times 300-1000. Under i), a protein with pLDDT x in (80, 90] will be used for times t>=300 * (90 - x) / (90 - 80), e.g. a protein with pLDDT 82 is used for times t>=240.
>
> | `plddt_to_timestep`| Designability | Diversity | Pareto         | Notes        |
> |-----------------------------|---------------|-----------|----------------|--------------|
> | (90,0),(80,300),(70,900)    | **98.6**      | 0.781     | ✅ Pareto optimal | Paper choice |
> | Piecewise linear            | 96.2          | 0.827     | ✅ Pareto optimal |              |
> | Sigmoid                     | 98.2          | 0.752     |                |              |
> | (90,0),(85,300),(80,600),(75,900),(70,950) | 95.9 | 0.742 |                | Extra bins 1 |
> | (90,0),(85,200),(80,300),(75,850),(70,900) | 94.2 | **0.858** | ✅ Pareto optimal | Extra bins 2 |
>
> As shown, by optimizing the mapping function, we can even outperform the results we obtained in the paper. That said, even a simplistic choice, as the ones we opted for in the paper, can yield significant boosts.
>
> **Reviewer**: "Is there a principled way to optimize binning or the function?"
>
> Excellent question. Indeed, there is a principled way to find the mapping function, but implementing it might add some additional engineering steps. Essentially, the mapping function predicts the minimum amount of noise we need so that some distance (e.g., TV or KL) between the two distributions becomes less than epsilon. We can train a discriminator network to estimate this distance by seeing how much noise we need to add to confuse the discriminator. In our paper, we did not opt for this more principled solution because we were afraid it might hinder the wider adoption of the method, but we can certainly have this discussion for the Camera Ready.
>
> ### Secondary Structure Evals
>
> **Reviewer**: "have the authors examined generated structure topologies in more detail?"
>
> Following the Reviewer’s recommendation, we systematically evaluate the Secondary Structure Proportions of our generated proteins. Specifically, we first use biotite (as in Proteina) to assign secondary structure labels to all the generated proteins. Then, we look at the designable PDB files, we calculate the percentages of alpha/beta/loops, and then we average those over all designable PDBs. The results are shown below:
>
> | Method                                | α-Helix (%) ↓ | β-Sheet (%) ↑ |
> |---------------------------------------|---------------|----------------|
> | Genie2                                | 72.7          | 4.8            |
> | Proteina MFS 0.35                     | 71.6          | 5.8            |
> | Proteina MFS 0.45                     | 68.1          | 6.9            |
> | Proteina MFS 0.50                     | 67.0          | 7.2            |
> | Ambient Protein Diffusion (scale 0.5) | 69.9          | 7.9            |
> | Ambient Protein Diffusion (scale 0.4) | 66.6          | **9.5**        |
>
>
> As shown, Ambient Protein Diffusion generates proteins with more beta sheets and loops compared to Proteina and Genie2, illustrating progress towards designing more natural proteins.
>
> **Reviewer**: "What steps could mitigate the bias towards alpha helices?"
>
> Overall, with unconditional generation is really hard to fully capture the underlying distribution. This problem is known for image generative models too, e.g. in the recent NVIDIA paper “Guiding a diffusion model with a bad version of itself” the authors mention:
>
> “"Furthermore, the unconditional case tends to work so poorly that the corresponding quantitative numbers are hardly ever reported. The EDM2-S model trained with ImageNet-512, for example, yields a FID of 2.56 in the class-conditional setting and 11.67 in the unconditional setting."”
>
> The obvious solution is thus conditioning. Over the past two months (since the submission), we trained a model that can be conditioned on a per-residue SS label. As we increase the Classifier Free Guidance strength towards beta sheets, this model generates proteins with more interesting SS structure:
>
> | Strength Setting | α-Helix (%) ↓ | β-Sheet (%) ↑ |
> |------------------|---------------|----------------|
> | 0.2              | 82.1          | 1.5            |
> | 0.4              | 61.8          | 11.4           |
> | 0.6              | 30.3          | 33.6           |
> | 0.7              | 21.7          | **36.1**       |
>
> We plan to make these models available together with the code release upon acceptance of our work. We will further add all the Ablations and the Secondary Structure Evaluations in the paper.
>
> **Reviewer**: "This could limit the diversity of discovered proteins and undercuts one of the paper's main claims of achieving unprecedented diversity."
>
> Besides the SS evaluation results shown above, we further report novelty numbers for the models of the paper (in short and long generation) to truly show that our model can generate unique proteins:
>
> | Setting                   | Model                    | PDB Novelty ↓ | AFDB Novelty ↓ |
> |---------------------------|--------------------------|----------------|-----------------|
> | Long Protein Generation   | Ambient Protein Diffusion | **0.682**      | **0.740**       |
> |                           | Proteina                  | 0.836          | 0.883           |
> | Short Protein Generation  | Ambient Protein Diffusion | **0.774**      | **0.848**       |
> |                           | Proteina                  | 0.832          | 0.891           |
>
> **We believe that our rebuttal strongly addresses the Reviewer’s concerns, and we would be grateful if the Reviewer considers upgrading their score to reflect this.**

---

> > ### Author Response · Authors · 2025-08-08
> > **Request for feedback**
> >
> > Dear Reviewer,
> > As the deadline for the Authors-Reviewers discussion approaches, we would like to ask if our rebuttal has addressed your concerns. We put significant effort into the rebuttal, and we would really appreciate it if you could let us know if you found our answers satisfactory.
> >
> > Thank you for your time and service and for helping us improve our work!

---

> ### Comment · Area_Chair_SH6o · 2025-08-08
> **Please respond to reviewers updated response**
>
> Dear Reviewer,
>
> The end of the rebuttal is fast approaching. The authors would like to know if their new responses satisfy you further. I encourage you to please respond to their message at your earliest convenience.

---

### Official Review · Reviewer_6ytr · 2025-07-06

**Clarity:** 3
**Significance:** 3
**Originality:** 3
**Rating:** 5
**Confidence:** 4

**Summary:**

The paper proposes Ambient Protein Diffusion, which trains diffusion models on noisy AlphaFold structures by adjusting the objective based on corruption levels, avoiding filtering low-pLDDT data. It reclusters AFDB for geometric diversity, boosting diversity to 85% and designability to 88% for 700-residue proteins, outperforming prior work with a 16.7M-parameter model. This advances de novo protein design for complex lengths by leveraging both high and low-quality structures.

**Questions:**

- The spirit of the ambient diffusion is to utilize low quality data where only timesteps t>ti are taken into consideration. I am not sure whether first construciting the (x(t(i)), t(i)) paris is necessraty. Since you can just first sample a data point, and sample t according to its plddt. In this way, you do not need to change the loss funciton (parameteerized by x(t(i)), and using the original loss function (h(xt) - x0) is enough. I think they are mathematically equvilant. Correct my if I am misunderstanding the approach.
- In line 318, did you remove the low-quality data?

**Ethical Concerns:**

["NO or VERY MINOR ethics concerns only"]

**Final Justification:**

My questions are fully resolved. This is a technically solid paper.

**Quality:**

3

**Strengths And Weaknesses:**

## Strengths

- This study fully leverages low-pLDDT AlphaFold structures by treating them as “corrupted” data, a novel approach in protein design that avoids discarding valuable low-confidence structures and instead adjusts diffusion objectives based on corruption levels.
- The method achieves SOTA performance on long proteins (up to 700 residues), boosting diversity from 45% to 85% and designability from 70% to 88% compared to prior methods, with a significantly smaller model (16.7M parameters).
- This study constructs a new training dataset by reclustering AFDB for geometric diversity (not jsut evolutionary similarity), yielding structurally diverse clusters that better represent protein structure space for generative modeling.



## Weaknesses

- Limited ablation studies on some key parameters.
  - How does the mapping function (f) (line 202) influence the results?
  - Can you present diffirent choices of the weighting factor w(t) (line 215)?
- Unclear contribution attribution: No systematic ablations separate the effects of the geometrically reclustered dataset from the low-pLDDT utilization method, making it difficult to determine which component drives performance gains. (besides this concern, I believe that the ambient diffusion works.)
- The pLDDT threshold: Since you have adopted ambient diffusion, why do you filter structures with pLDDT <70 (line 294)? This potentially excludes valuable low-confidence data.
- Model comparison: Training distinct models for short and long proteins (line 268) and comparing with single-model baselines could be a bit strange.
- In Figure 5, the ablation plot lacks a direct comparison with Proteína, which performs well on long proteins.

---

> ### Author Rebuttal · Authors · 2025-07-31
>
> We thank the Reviewer for their time and feedback. In what follows, we present numerous ablations. We report (designability, diversity) pairs for the short-protein generation benchmark (Figure 4), as it is computationally infeasible to do long protein training for all these studies.
>
> ### Mapping Function Ablations
>
> **Reviewer**: "How does the mapping function influence the results?"
>
> **Great question – we thank the Reviewer for raising this.**
>
> #### Robustness
>
> In the paper, we used the function `f` defined by the `plddt_to_timestep` mapping:
>
> (90, 0), (80, 300), (70, 900)
>
> Proteins with:
> - **pLDDT > 90** are used throughout all steps (0 to 1000).
> - **pLDDT in (80, 90]** are used from step 300 onward.
> - **pLDDT in (70, 80]** are used only from step 900 onward.
>
> We will now show what happens when we slightly change the function f we used in the paper in two opposite directions.
>
> | `plddt_to_timestep` | Designability | Diversity | Notes |
> |---------------------|---------------|-----------|-------|
> | (80,0) | 95.2 | 0.590 | Genie2 baseline |
> | **(90,0),(80,300),(70,900)** | **98.6** | 0.781 | Paper choice |
> | (85,0),(75,300),(70,900) | 96.4 | 0.780 | Deflated thresholds by 5 |
> | (95,0),(85,300),(75,900) | 96.8 | **0.783** | Inflated thresholds by 5 |
>
>
> As shown, our method is relatively robust, as small perturbations to the mapping don’t lead to a very substantial change. In fact, the inflated thresholds lead to a model that belongs to the Pareto frontier of Figure 4 in the paper. Running the same model with scale=0.65 leads to another Pareto point, achieving an (86.6, 0.882) designability-diversity pair.
>
> #### Continuous mapping functions and more bins
>
> We further present continuous generalizations of our original choice. In the paper, we chose a very rough categorization of the proteins into “high-quality”, medium-quality, and “low-quality” for didactic purposes (L191-200). We now try:
>
> i) a piecewise linear continuous generalization.
> ii) a sigmoid.
> iii) more bins
>
> E.g. before all proteins with pLDDT in (80, 90] were used for times 300-1000. Under i), a protein with pLDDT x in (80, 90] will be used for times t>=300 * (90 - x) / (90 - 80), e.g. a protein with pLDDT 82 is used for times t>=240.
>
> | `plddt_to_timestep`| Designability | Diversity | Pareto         | Notes        |
> |-----------------------------|---------------|-----------|----------------|--------------|
> | (90,0),(80,300),(70,900)    | **98.6**      | 0.781     | ✅ Pareto optimal | Paper choice |
> | Piecewise linear            | 96.2          | 0.827     | ✅ Pareto optimal |              |
> | Sigmoid                     | 98.2          | 0.752     |                |              |
> | (90,0),(85,300),(80,600),(75,900),(70,950) | 95.9 | 0.742 |                | Extra bins 1 |
> | (90,0),(85,200),(80,300),(75,850),(70,900) | 94.2 | **0.858** | ✅ Pareto optimal | Extra bins 2 |
>
> As shown, by optimizing the mapping function, we can even outperform the results we obtained in the paper. That said, even a simplistic choice, as the ones we opted for in the paper, can yield significant boosts.
>
> ### Weight Ablations
>
> **Reviewer**: "Can you present diffirent choices of the weighting factor w(t)?"
>
> We ablate a constant weighting function (w=1) as used in the Genie2 paper and the square root of the weighting used in the paper.
>
> | Weighting         | Designability | Diversity | Pareto         |
> |--------------------------|---------------|-----------|----------------|
> | Paper choice             | **98.6**      | 0.781     | ✅ Pareto optimal |
> | Constant (w=1)           | 97.4          | 0.733     |                |
> | Square root              | 95.0          | **0.783** |                |
>
> As shown, the weighting we picked in the paper yields the **optimal results**. The reason this reweighting is needed is due to the factor multiplying the network prediction in the loss of Eq. (2). The reweighting we proposed corrects for this multiplication, following the derivations in the paper:
> **Elucidating the Design Space of Diffusion Models (EDM)**
> — *Appendix, p. 27, Section B6*.
>
> We will detail the derivation in the Camera Ready version of our work.
>
> #### Figure 5 clarifications
>
> **Reviewer:** "In line 318, did you remove the low-quality data?"
>
> Figure 5 of our paper compares **three models**:
>
> 1. **Original Genie2**
>    Trained on short proteins with the original Genie2 dataset (pLDDT > 80 filtering, no reclustering).
>
> 2. **Improved Genie2 (Baseline w/o Ambient)**
>    Trained on longer proteins using the **re-clustered data, filtered at pLDDT > 70**.
>
> 3. **Ambient Diffusion**
>    Trained on longer proteins using the **re-clustered data** filtered at pLDDT > 70,
>    *plus* the Ambient methodology for utilizing low-quality pLDDT samples.
>
> In summary, this figure **fixes the dataset** (re-clustered, pLDDT > 70) and **ablates the Ambient methodology**.
>
> **Reviewer**: "In Figure 5, the ablation plot lacks a direct comparison with Proteína, which performs well on long proteins."
>
> The comparison (designability, diversity) is shown below:
>
> | Length | Original Genie2 | Improved Genie2 | Proteina     | Ambient Protein Diffusion |
> |--------|------------------|------------------|--------------|----------------------------|
> | L300   | (81,90)          | (93,93)          | (93,55)      | (90,89)                    |
> | L400   | (60,60)          | (79,79)          | (85,60)      | (92,92)                    |
> | L500   | (20,20)          | (71,71)          | (82,65)      | (91,91)                    |
> | L600   | (2,3)            | (51,51)          | (81,55)      | (87,87)                    |
> | L700   | (1,0)            | (30,30)          | (68,45)      | (86,86)                    |
> | L800   | (0,0)            | (25,25)          | (55,47)      | (68,68)                    |
>
> As shown, **our Ambient Protein Diffusion model massively outperforms Proteina in long protein generation.**
>
> ### More Dataset & Ambient Ablations
>
> **Reviewer**:  "No systematic ablations separate the effects of the dataset from the low-pLDDT utilization method"
>
> Figure 5 in the paper directly ablates the effect of using Ambient Diffusion, as it shows the effect of having it and not having it on the same exact dataset (reclustered filtered at pLDDT > 70).
>
> We agree with the Reviewer that we need to separately ablate the need for having a re-clustered dataset. Results below:
>
> | Setting                     | Dataset              | Designability ↑ | Diversity ↑ |
> |----------------------------|----------------------|------------------|--------------|
> | No Ambient                 | Genie2      | 95.2             | 0.590        |
> | No Ambient                 | Re-clustered         | 81.4             | 0.902        |
> | Ambient                    | Genie2      | 96.4             | 0.501        |
> | Ambient                    | Re-clustered         | **98.6**         | **0.781**    |
>
> **Reclustering leads to improved diversity**.
>
> ### Other clarifications
>
> **Reviewer**: "why do you filter structures with pLDDT <70?"
>
> It is true that we could use proteins with lower pLDDTs too. That said, proteins with pLDDT ~70, are already used for only 10% of the diffusion training (steps from 900-1000). Hence, we don’t expect a dramatic improvement by using proteins with pLDDT<70 with our method, and we skipped those to avoid the whole AFDB dataset (estimated size ~28 terabytes).
>
> **Reviewer**: " Training distinct models for short and long proteins [...] could be a bit strange"
>
> There seems to be a misconception here. Our Genie2 baseline is only trained (and optimized) for short protein generation. Proteina also uses different models to produce their short-generation and long-generation results. In fact, in Proteina there are four different models trained for short generation (Table 1 from https://arxiv.org/pdf/2503.00710). Proteina finetunes one of the four short generation models (the MFS^{no-tri}) for long generation and reports results in Figure 8 of their paper. Similarly, we fine-tune the short-protein generation model for long generation, and we report results in Figure 3 of our submission.
>
>
> **Reviewer**: "[...] you do not need to change the loss function [...]"
>
> Excellent question – let us clarify it here.
>
> The Reviewer proposes predicting x0. For low pLDDT structures, x0 cannot be trusted. Instead, our loss involves predicting x_{t_n}, which can be trusted more than x0 because the noise has erased some of the AlphaFold prediction mistakes. So in terms of loss, the prediction of x_{t_n} instead of x_0 is actually critical. For completeness and to support our argument, we did one more ablation during the rebuttal where we trained a model with the same setup as the short-protein generation in the paper, but using x0 prediction loss instead of the proposed loss. Results:
>
> | Loss Type           | Designability | Diversity |
> |---------------------|----------------|-----------|
> | Ambient loss        | **98.6**       | **0.781** |
> | x₀ prediction loss  | 98.5           | 0.753     |
>
> Moving on to the Reviewer’s point about the sampling of times: is it necessary to first noise to $x_{t_n}$ and then to $x_t$ instead of noising all the way directly?
>
> The answer is surprisingly yes (as we hint in the paper L208-209 and in our pseudocode Algorithm 1, Appendix page 6). The reason has to do with the information leakage about x0 that happens as we do multiple epochs. When the corruption to noise level x_t_n happens once, as we do in our paper, x0 cannot be fully recovered – some part of the signal has been destroyed by noise. However, if we are allowed to get multiple noise versions of it at the same noise level then we can average them out and approximately recover the signal. That’s why in a multiple epochs setting the two ways of getting x_t are not equivalent.
>
> **We believe that our rebuttal strongly addresses the Reviewer’s concerns, and we would be grateful if the Reviewer considers upgrading their score to reflect this.**

---

> > ### Comment · Reviewer_6ytr · 2025-08-01
> >
> > Thank you for your effort to resolve most of my concerns. I am happy to raise my score.
> >
> > I still have some questions:
> >
> > - Regarding the last question, although the authors conduct further experiments to show that the ambient loss is better than the $x_0$ loss, I am curious about whether this is more like the difference between $\epsilon$-parameterization and $x_0$-parameterization in traditional diffusion training -- I mean, are they theoretically equivariant, although performance may be different in practice?
> >
> > - What do you mean by *When the corruption to noise level x_t_n happens once, as we do in our paper, x0 cannot be fully recovered*? I think for all diffusion models, $x_0$ can not be fully recovered.

---

> > > ### Author Response · Authors · 2025-08-02
> > > **Thank you and clarification about Ambient Loss**
> > >
> > > We thank the Reviewer for reading our rebuttal and for engaging with us. We are very pleased to hear that most of the concerns are resolved and that the Reviewer will increase their score. The Reviewer raises some meaningful questions regarding the need for Ambient Loss.
> > >
> > > **Reviewer**: "are they theoretically equivariant, although performance may be different in practice?"
> > >
> > > In the finite samples setting, **the two losses are not equivalent**. The two losses become equivalent when we have access to infinite samples. There is a paper called "Does Generation Require Memorization? Creative Diffusion Models using Ambient Diffusion" (ICML 2025) that actually delves into the differences of the Ambient Loss vs the x0 prediction loss for the finite samples setting. We point the Reviewer to this work if the Reviewer wants a deep dive into the topic. However, we do our best to clarify below (and we will add the relevant discussion to the main paper too).
> > >
> > > The idea is that when we train our diffusion models, we have access to the empirical distribution $p_0$ that can be viewed as a bunch of diracs at the training points. If a model is trained under the x0 prediction loss and it manages to minimize perfectly its training objective (i.e. if it learns the conditional expectation for the empirical distribution) the model will perfectly reproduce the training samples at inference time and nothing else. This observation was made in the papers "Closed Form Diffusion Models" and "An analytic theory of creativity in convolutional diffusion models". A generative model that only reproduces the training samples is not very useful. Ideally, we would like to sample from the underlying continuous distribution of proteins that led to the observed empirical samples. Thankfully, in practice many times we fail to perfectly optimize the training objective (due to limited epochs, architectural/optimization biases, etc) and hence creativity arises (see An analytic theory of creativity in convolutional diffusion models). That said, oftentimes diffusion models trained with x0-prediction losses or other equivalent parametrizations perfectly memorize some of their training samples and such samples can be extracted at inference time (e.g. see the works: "Extracting Training Data from Diffusion Models" and "Consistent Diffusion Meets Tweedie").
> > >
> > > With the Ambient Loss, it is impossible to ever perfectly memorize a (low-quality) dataset sample because, prior to training, these samples are getting noised to a noise level $x_{t_n}$. After this first noising happens, the underlying sample $x_0$ is completely discarded (i.e. it is like as it never existed). The training only uses the single realization of the random variable $X_{t_n}$ to produce extra corrupted samples at noise levels $t \geq t_n$. Since the corruption happens once at the beginning of the training and $x_0$ is not perfectly recoverable from a single realization of $x_{t_n}$, the Ambient Loss induces a different population minimizer than the x0-prediction loss. The losses become equivalent only if we have infinite samples. For more details, we point the Reviewer to Lemmas 4.1 and 4.2 of the ICML paper: "Does Generation Require Memorization".
> > >
> > >
> > > **Reviewer**: "What do you mean by When the corruption to noise level x_t_n happens once, as we do in our paper, x0 cannot be fully recovered? I think for all diffusion models, x0 can not be fully recovered."
> > >
> > > Maybe the discussion above already clarifies this point. A diffusion model at sampling time can sometimes perfectly reproduce a training sample (e.g. see the works: "Extracting Training Data from Diffusion Models", "Consistent Diffusion Meets Tweedie", "Diffusion art or digital forgery? investigating data replication in diffusion models"). However, low-quality samples that are used via the Ambient Loss cannot be perfectly recovered because there is information loss about $x_0$ as we discard the sample and we replace it with a noisy version of itself $x_{t_n}$ before the training starts.
> > >
> > >
> > > We hope the Reviewer's justified question is now resolved. If not, we remain at the Reviewer's availability for futher clarifications. We commit to making this point more clear in the Camera Ready version -- including a more detailed discussion of the paper "Does Generation Require Memorization" that provides meaningful theoretical analysis of the two losses.

---

> > > > ### Comment · Reviewer_6ytr · 2025-08-03
> > > >
> > > > Thank you for your response. I have no further questions!

---

> > > > > ### Author Response · Authors · 2025-08-03
> > > > > **Thank you!**
> > > > >
> > > > > Wonderful! Thank you for your time and feedback. We will make sure to include our additional experiments and the discussion in the Camera Ready version of our work.

---

### Decision · Program_Chairs · 2025-09-17

**Decision:**

Accept (spotlight)

**Comment:**

This paper introduces Ambient diffusion for proteins. All reviewers agree that the data engineering and new loss function lead to robust gains across in silico benchmarks. There were several questions regarding ablations of the map $f$ as well as understanding secondary structure diversity. The authors have answered point 1.) with new experiments, and shown that with guidance $\beta$-sheet diversity can be improved. As a result, the use of Ambient diffusion can be leveraged for all future protein diffusion models. However, it is unclear to what extent the in silico gains will be reflective of actual wet lab performance, as AFDB is itself inherently biased. Despite this, in the context of NeurIPS I recommend acceptance with Spotlight.